# Atlas of multilineage stem cell differentiation reveals *TMEM88* as a developmental regulator of blood pressure

Sophie Shen [1], Tessa Werner[1], Samuel W. Lukowski[1], Stacey Andersen[1,2], Yuliangzi Sun[1], Woo Jun Shim [1], Dalia Mizikovsky[1], Sakurako Kobayashi [1], Jennifer Outhwaite[1], Han Sheng Chiu[1], Xiaoli Chen[1], Gavin Chapman[3,4], Ella M. M. A. Martin [3], Di Xia[1,2], Duy Pham[1], Zezhuo Su [5,6], Daniel Kim[7], Pengyi Yang[7,8], Men Chee Tan[1,9], Enakshi Sinniah[1], Qiongyi Zhao [1], Sumedha Negi [1], Meredith A. Redd[1], Joseph E. Powell [10,11], Sally L. Dunwoodie [3,4,12], Patrick P. L. Tam [13,14], Mikael Bodén [15], Joshua W. K. Ho [5,6], Quan Nguyen [1] & Nathan J. Palpant [1,8] ✉

Pluripotent stem cells provide a scalable approach to analyse molecular regulation of cell differentiation across developmental lineages. Here, we engineer barcoded induced pluripotent stem cells to generate an atlas of multilineage differentiation from pluripotency, encompassing an eight-day time course with modulation of WNT, BMP, and VEGF signalling pathways. Annotation of in vitro cell types with reference to in vivo development reveals diverse mesendoderm lineage cell types including lateral plate and paraxial mesoderm, neural crest, and primitive gut. Interrogation of temporal and signalling-specific gene expression in this atlas, evaluated against cell type-specific gene expression in human complex trait data highlights the WNT-inhibitor gene *TMEM88* as a regulator of mesendodermal lineages influencing cardiovascular and anthropometric traits. Genetic *TMEM88* loss of function models show impaired differentiation of endodermal and mesodermal derivatives in vitro and dysregulated arterial blood pressure in vivo. Together, this study provides an atlas of multilineage stem cell differentiation and analysis pipelines to dissect genetic determinants of mammalian developmental physiology.

Human pluripotent stem cell (hPSC) models are ideally suited to deconstruct early human developmental processes in a scalable and cell type-specific manner. Significant investment and sector growth in stem cell technologies underscore their importance in developmental biology research, industry drug discovery pipelines, disease modelling, and regenerative medicine[1]. Multiplexed single-cell omics technologies offer a means to massively upscale data generation to enable these applications, but while these technologies have been broadly applied to characterise in vivo biology[2-4], atlases of in vitro hPSC differentiation are not widely available. As a consequence, mechanisms governing hPSC differentiation and fate specification remain inadequately understood and characterised[5-7], creating barriers in guiding the differentiation of hPSCs into accurate models of development and disease[8-10]. Therefore, strategies are needed to efficiently characterise the sequential stages of hPSC differentiation to maximise relevance of hPSC-derived cell types for applications in research and beyond.

Current sample multiplexing methods for single-cell experiments either require samples with different genetic backgrounds[11,12] or involve

---

additional cell handling to apply sample barcodes prior to sequencing[13–15], both of which introduce technical variation to the data readout. Additionally, translating insights from in vitro hPSCs into physiological mechanisms of development and disease remains challenging due to the lack of complex tissue and organism-level phenotypes. Genome-wide association studies (GWAS) and databases mapping genetic diversity to transcriptional variation across tissues (GTEx) present interesting avenues for understanding how genetic changes observed in vitro can influence phenotypic variation in human populations.

In this study, we develop a multiplexing method for single-cell RNA sequencing (scRNA-seq), enabled by genomic integration of transcribed barcodes in human induced pluripotent stem cells (iPSCs). Using these barcoded iPSCs, we perform a multiplexed perturbation study to characterise multilineage diversification of cells from pluripotency in vitro. We analyse this in vitro differentiation atlas using a suite of computational methods to identify regulators of temporal and signalling-specific cell fate specification. By evaluating genetic regulators of in vitro cell diversification against tissue-specific gene-trait associations derived from GWAS and GTEx data, we identify and validate transmembrane protein 88 (TMEM88), as a regulator of cardiovascular development with impacts on blood pressure phenotypes in adult mice. Collectively, this study provides cell-based experimental tools, data resources, and computational workflows that leverage the power of atlas-level multilineage in vitro cell diversification to discover genetic determinants of developmental physiology.

## Results

### Derivation of barcoded iPSCs for multiplexed single-cell RNA-seq

Cell barcoding strategies enable sample multiplexing and economise the effort and costs of data generation by scRNA-seq analysis. To enhance sample multiplexing capabilities in human induced pluripotent stem cells (hiPSCs), we designed a cell barcoding method to enable multiplexed single-cell analysis of isogenic human cell lines (Fig. 1a). 10,000 15-base pair barcodes were randomly generated using an even 25% probability for the presence of each A, C, T and G nucleotides. After excluding barcodes starting or ending with a stop codon or containing repetitive runs of 4 or more nucleotides, we selected 18 barcodes, requiring a minimum Hamming distance of 5 nucleotides between each pair of barcodes (Table 1). Using this design, we built barcode plasmids into the AAVS1-2A-Puro gene targeting cassette[16,17] to enable expression of a barcoded GFP transcript downstream of the ubiquitously expressed CAG at the human *AAVS1* locus after CRISPR-Cas9 integration (Supplementary Data 1). Transcribed barcodes can thus be captured by any single-cell RNA-seq protocol and amplified in a separate sequencing library to allow each cell from pooled experimental samples to be assigned back to its sample of origin without the need for exogenous labelling before sample pooling (Fig. 1a). With thousands of potential barcode combinations, the extent of multiplexing is only limited by the scale of cell capture technologies. We engineered 18 isogenic hiPSC lines with unique barcodes, each of which passed quality testing for karyotype, pluripotency, and purity (Fig. 1b–e).

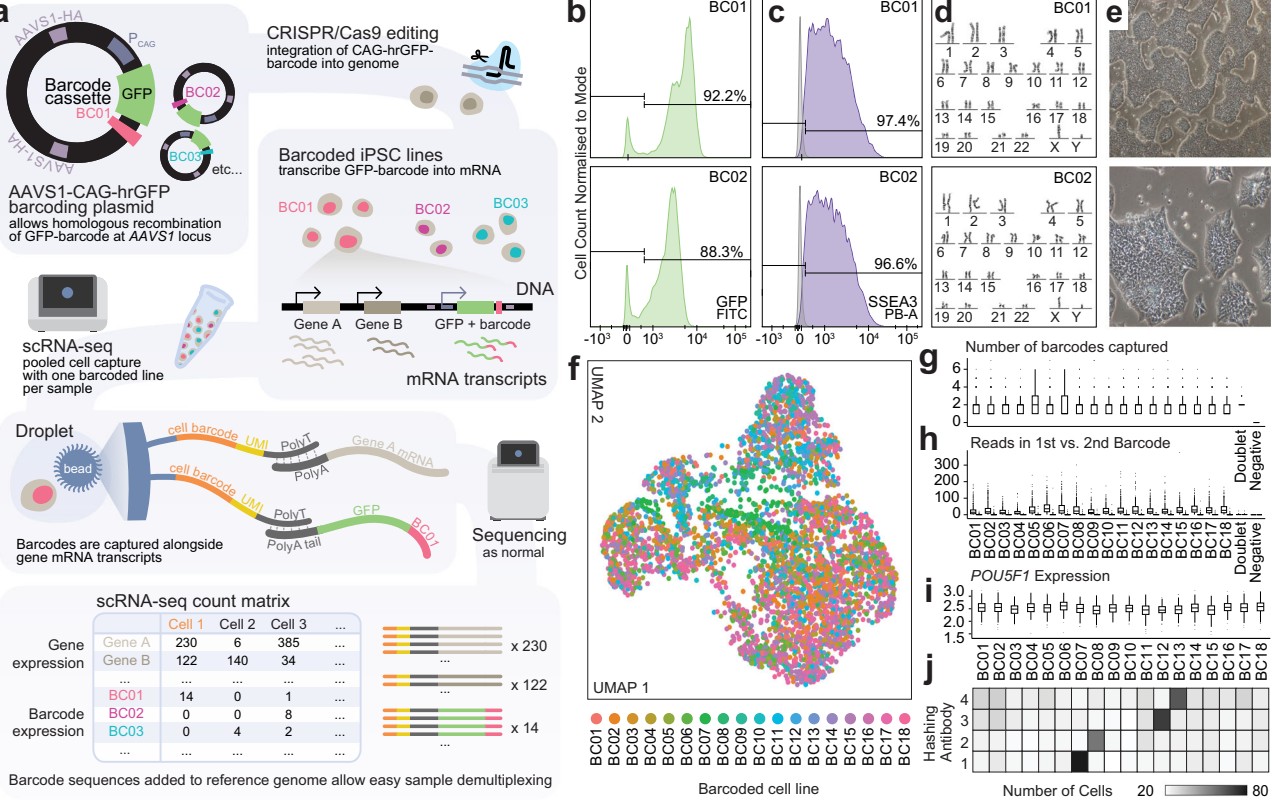

Fig. 1 | **Engineered barcoding of human pluripotent stem cells. a** Workflow for CRISPR/Cas9 engineering of barcoded hiPS cell lines to facilitate sample multiplexing of isogenic samples in scRNA-seq experiments. AAVS1-HA: *AAVS1* Homology Arm; hrGFP: humanised Renilla Green Fluorescent Protein; P_CAG: CAG promoter. See also "**Methods**" and Supplementary File 1. **b–e** Quality control for barcoded cell lines including representative FACS plots for GPF+ (vector integration) (**b**) and SSEA3+ (pluripotent) cells (**c**), karyotype (**d**), and images of pluripotent cell morphology (**e**) for barcoded cell lines BC01 and BC02. **f** UMAP of pilot scRNA-seq experiment capturing all 18 barcoded cell lines as undifferentiated iPSCs. Points are coloured by the barcode with the highest number of mapped reads per cell. **g–i** Box plots showing number of unique expressed genomic barcodes (**g**), reads mapped to the top two most highly mapped barcodes (**h**) and expression level of pluripotency gene *POU5F1* (OCT4) (**i**) in each cell, grouped by the dominant barcode in each cell. Centre line shows median, box limits show upper and lower quartiles, whiskers show 1.5x interquartile range, and points show outliers. See also Supplementary Fig. 1a and Supplementary Table 1 for more metrics and n for each group. **j** Independent validation of barcoding by analysing barcoded cell lines co-labelled with external cell hashing antibodies. Antibodies 1-4 were stained to cells in barcoding lines BC07, BC08, BC12, and BC13 respectively. Source data are provided as a Source data file.

**Table 1 | Barcodes and multiplexed experimental design**

| Barcode Sequence | Line | Condition | Library 1 | Library 2 | Library 3 |
|---|---|---|---|---|---|
| GTGCCGACCAGTATC | BC01 | Control | Day 5 | Day 2 | Day 9 |
| ACCACCTGACGCAAA | BC02 | 1 µM XAV939 | Day 5 | Day 2 | Day 9 |
| ACGGCCCTATTTAAG | BC03 | 5 µM XAV939 | Day 5 | -- | Day 9 |
| AGCCCTGAGTCAGTA | BC04 | 3 µM CHIR99021 | Day 5 | Day 2 | Day 9 |
| CAAATTCAAGGCGAT | BC05 | 0.4 µM Dorsomorphin | Day 5 | Day 2 | Day 9 |
| AATCTTGTATAAGTA | BC06 | 4 µM Dorsomorphin | Day 5 | Day 2 | Day 9 |
| CGTCACATTTGAGTC | BC07 | 0.5 µM K02288 | Day 5 | Day 2 | Day 9 |
| GGACCTTCTTACGAC | BC08 | 10 ng/ml BMP4 | Day 5 | Day 2 | Day 9 |
| ACCCTACGGTGGTTC | BC11 | 100 ng/ml VEGF | Day 5 | Day 2 | Day 9 |
| CGCTAATGTCCGTTT | BC10 | Control | Day 2 | Day 9 | Day 5 |
| TACCAATTGTACGCT | BC09 | 1 µM XAV939 | Day 2 | Day 9 | Day 5 |
| TGTCCAAGCTGCAAT | BC12 | 5 µM XAV939 | Day 2 | Day 9 | Day 5 |
| GTGTATTTAAAGCCG | BC13 | 3 µM CHIR99021 | Day 2 | Day 9 | Day 5 |
| ACACCCGTATGTCAC | BC14 | 0.4 µM Dorsomorphin | Day 2 | Day 9 | Day 5 |
| TCTTTCGATGGCGGT | BC15 | 4 µM Dorsomorphin | Day 2 | Day 9 | Day 5 |
| GAGCACCCGCGTATT | BC16 | 0.5 µM K02288 | Day 2 | Day 9 | Day 5 |
| TTATTATGTTCTAGC | BC17 | 10 ng/ml BMP4 | Day 2 | Day 9 | Day 5 |
| AATCTCTGAAACGAA | BC18 | 100 ng/ml VEGF | Day 2 | Day 9 | Day 5 |

We performed a pilot scRNA-seq experiment pooling all 18 barcoding lines as undifferentiated iPSCs to validate homogeneity of their transcriptomic profiles and to confirm that the transcribed barcodes could be used to easily assign a sample of origin for each cell (Fig. 1f). From a total of 6,711 cells captured, 19.4% had no reads mapped to any genomic barcode, however 95.0% of these cells with no genomic barcode reads also did not pass quality control on the basis of their low total read count, gene count, and high mitochondrial RNA content (Supplementary Table 1 and Supplementary Fig. 1a). The remainder of cells had reads mapped to between one and seven genomic barcodes, where on average, cells were mapped to two barcodes, with just one read mapped to the lower "expressed" barcode (Supplementary Table 1 and Fig. 1g, h). Dimensionality reduction analysis shows that transcriptomes are sufficiently homogenous to randomly distribute across UMAP space (Fig. 1f), and cells from each line reveal comparable reads per cell and expression of pluripotency markers *(POU5F1)* (OCT4) and *SOX2* (Fig. 1i and Supplementary Fig. 1a). We additionally used cell hashing antibodies[15] to label four samples (BC07, BC08, BC12, BC13) to benchmark the accuracy of our method against an established technology. 882 cells were mapped to this subset of genomic barcodes, while 1104 cells had cell hashing (HTO) reads mapped, with an 83.2% agreement between the genomic barcoding and cell hashing sample assignment (Fig. 1j and Supplementary Table 1). We also observed comparable transcriptomic quality between the two methods (Fig. 1g–j and Supplementary Fig. 1a), providing support for the reliability of using genomic transcribed barcodes for isogenic sample multiplexing in scRNA-seq experiments.

**Generation of a multiplexed atlas of iPSC differentiation in vitro**
To further evaluate efficacy of these barcoded cell lines for sample multiplexing, we set out to utilise them to capture a dataset that could make full use of the multiplexing potential. Given the diverse signalling and molecular cues shown to inform cell lineage specification during early development, we generated a dataset that could be used to study the role of these molecular determinants during WNT-induced hiPSC differentiation from pluripotency[18,19] (Fig. 2a). First, we utilised cell hashing to capture a reference time course of differentiation under control conditions, collecting cells at eight time points; every 24 h from day 2 to day 9 of differentiation. Next, we treated differentiating cells between day 3 (germ layer differentiation) and day 5 (progenitor

cell stage) with perturbations including inhibitors and activators of WNT[20], BMP[21], and VEGF[22] signalling pathways known to be essential regulators of developmental lineage differentiation (Fig. 2b and Table 1). In this signalling perturbation dataset, cells were sampled in duplicates prior to perturbation (day 2), following perturbation (day 5), and at the differentiation endpoint of committed cell types (day 9).

In total, 62 independent cell samples were collected across four scRNA-seq libraries with comparable quality control metrics (Supplementary Table 2). Both the cell hashing library for the time course dataset and genomic barcoding libraries for the signalling perturbation dataset had greater sequencing depth, averaging 600 reads per cell in the former and 190 reads per cell in the latter libraries, where majority of cells had reads mapping to all possible cell hashing or genomic barcode labels (Supplementary Table 2 and Supplementary Fig. 1b). The cell hashing library showed a greater difference in the percentage of reads mapping to the first and second-highest expressed barcode per cell, however the multiplexing doublet and singlet rates as determined by conventional sample demultiplexing strategies were comparable between the two methods (Supplementary Table 2 and Supplementary Fig. 1b). Standard quality control and preprocessing of data were performed (Supplementary Fig. 1c), resulting in 13,682 high quality cells from the time course reference and 48,526 cells from the signalling perturbation dataset (Supplementary Fig. 1d–g).

The two datasets were integrated using established methods[23] into an atlas of multilineage hiPSC differentiation (Fig. 2c). Clustering was performed on the integrated dataset to reveal 13 cell clusters, whose marker gene expression and associated enriched GO terms and KEGG pathways were evaluated to reveal capture of mesendoderm (clusters 0 & 1; *MIXL1*), definitive endoderm (clusters 2 & 3; *FOXA2*) anterior (cluster 8; *SOX2*) and posterior (cluster 5; *TTR*) foregut endoderm progenitors, lateral plate (cluster 6; *HAND1*) and paraxial mesoderm (cluster 7; *PRRX1*), cardiomyocytes (cluster 10; *MYH6*), endothelium (cluster 12; *CDH5*), axial mesoderm (cluster 4; *NOG*), ciliated node cells (cluster 11; *FOXJ1*), and neural progenitors (cluster 9; *SOX2*) (Fig. 2d, e and Supplementary Fig. 2a–c). Further, expression of pluripotency marker *NANOG* in cluster 0 and enrichment of cell division-related GO terms in cluster 2 distinguish them as proliferative mesendoderm and definitive endoderm respectively (Supplementary Fig. 2a–c). Label transfer analysis with reference to mouse[2,24] and

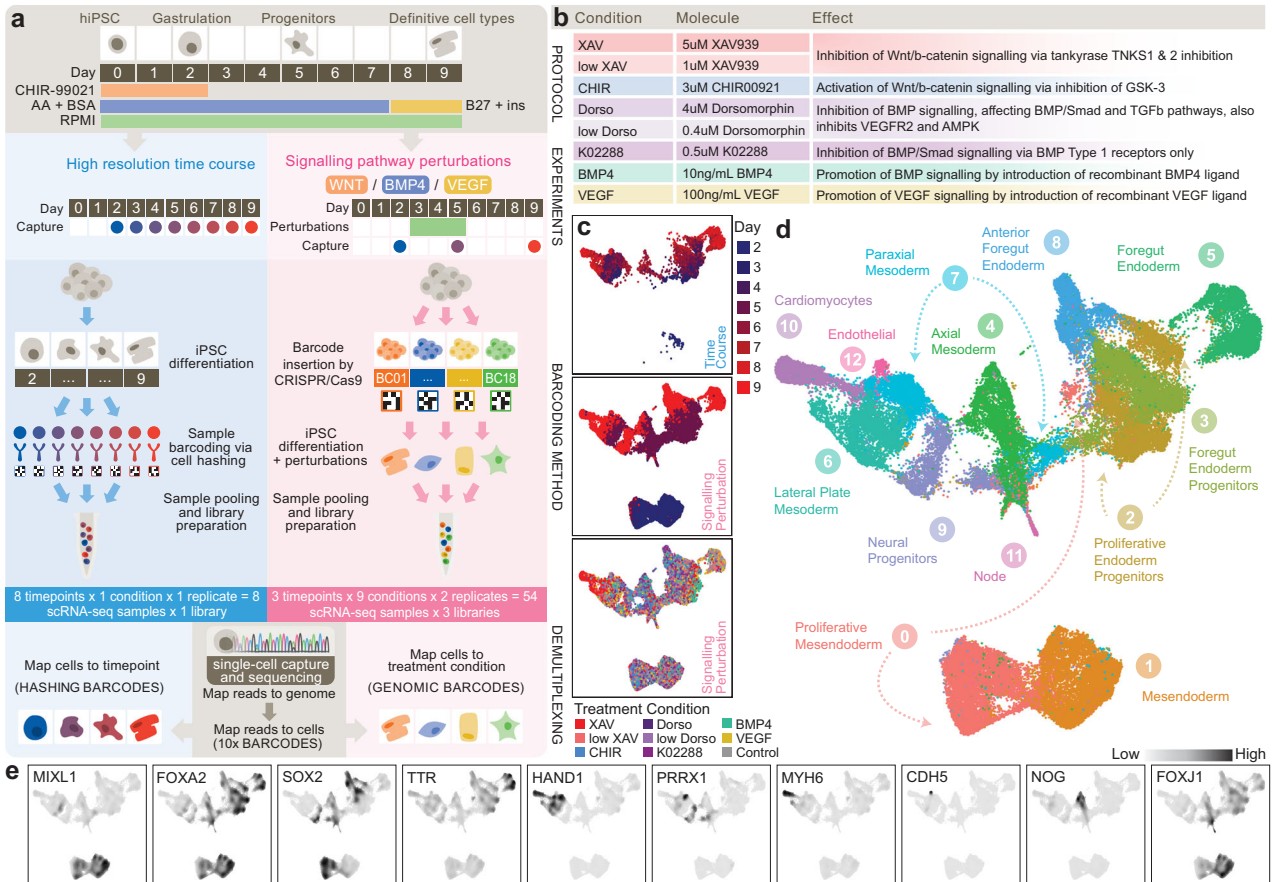

**Fig. 2 | Generating an integrated multilineage atlas of in vitro iPSC differentiation. a** Multiplexed study design including 1) a high-resolution time course dataset (blue, left) capturing cells every 24 h between days 2 and 9 of differentiation and 2) a signalling pathway perturbation dataset (pink, right) generated via perturbation of developmental signalling pathways between days 3 and 5 of differentiation with cells captured on days 2, 5, and 9. In total, 62 cell samples were captured and sequenced using 4 single-cell libraries. Samples were demultiplexed by reference to cell and sample barcodes. **b** Signalling perturbation conditions, concentrations, and mechanism of action. **c** UMAP plots of integrated dataset comprising 62,208 single cells, split by dataset of origin, coloured by time point (upper and middle), and signalling perturbation treatment (lower). **d** Louvain clustering identifies 13 cell type clusters. Cell type annotations are defined based on expression of marker genes. **e** Nebulosa[74] plots showing expression distribution of cell type marker genes. *MIXL1* marks mesodermal cells; *FOXA2* marks definitive endoderm; *SOX2* marks early epiblast, neural progenitors, and foregut progenitors; *TTR* marks primitive foregut progenitors; *HAND1* marks precardiac mesoderm; *PRRX1* marks limb bud mesoderm; *MYH6* mark cardiomyocytes; *CDH5* marks endothelial cells; *NOG* marks axial mesoderm; and *FOXJ1* marks ciliated nodal cells. See also Supplementary Fig. 1f. Source data are provided as a Source data file.

human[4] embryogenesis scRNA-seq datasets also provided support for these cell type labels (Supplementary Fig. 2d). To provide further annotation of the atlas dataset, we used established downstream analysis methods to characterise cell-cell communication between cell types resulting from each treatment condition using CellChat[25] (Supplementary Fig. 2e and Supplementary Data 2–3), analysis of enriched gene regulons using pySCENIC[26,27] (Supplementary Fig. 2f and Supplementary Data 4), and URD-predicted[28] differentiation trajectories relevant to each cluster (Supplementary Fig. 2g). Together, this dataset resource presents a deep phenotypic analysis of a time-resolved molecular cell atlas in vitro embedded with information of the molecular mechanisms of temporal and signalling control of lineage diversification and cellular heterogeneity during multilineage differentiation of induced pluripotent stem cells.

### Determination of cell subtypes during iPSC differentiation

Following characterisation of the broad cell type groups comprising the dataset, we sought to understand changes in specific cell subtypes from each signalling perturbation. A common approach to this task is to increase the resolution, or granularity, of clustering to delineate smaller cell groups. The problem with this purely algorithm-driven approach is a lack of biological justification for a correct clustering

resolution or cluster size to select. Given the nuanced signalling cues and small temporal increments in our atlas dataset, we anticipated this cell type ambiguity to be particularly relevant such that increasing clustering resolution would result in many small, uninformative clusters, rather than capturing cell subtypes.

To define such a transcriptional signature marking distinct, more biologically robust cell types, we use a suite of tools built from the foundational TRIAGE method[29]. In brief, TRIAGE is a method that uses consortium-level epigenetic patterns to identify genes predicted to have highly cell type-specific regulatory functions. In this way, TRIAGE provides an independent biological reference point for defining cell states. This method was adapted for application in single-cell analysis pipelines in TRIAGE-Cluster and TRIAGE-ParseR to help identify cell populations and gene regulatory networks underpinning diverse cell types (Fig. 3a and Supplementary Fig. 3a).

First, we used TRIAGE-Cluster to identify informative cell subtypes within broader cell type clusters, eliminating the need to evaluate tens or hundreds of small, transcriptionally similar clusters (Fig. 3a, Supplementary Fig. 3b and Supplementary Data 5). Based on the clustering output (Fig. 3a), we took three approaches to evaluate in vitro-derived cell peaks by reference to in vivo derived developmental cell types. We first used gene set scoring to analyse cells based on

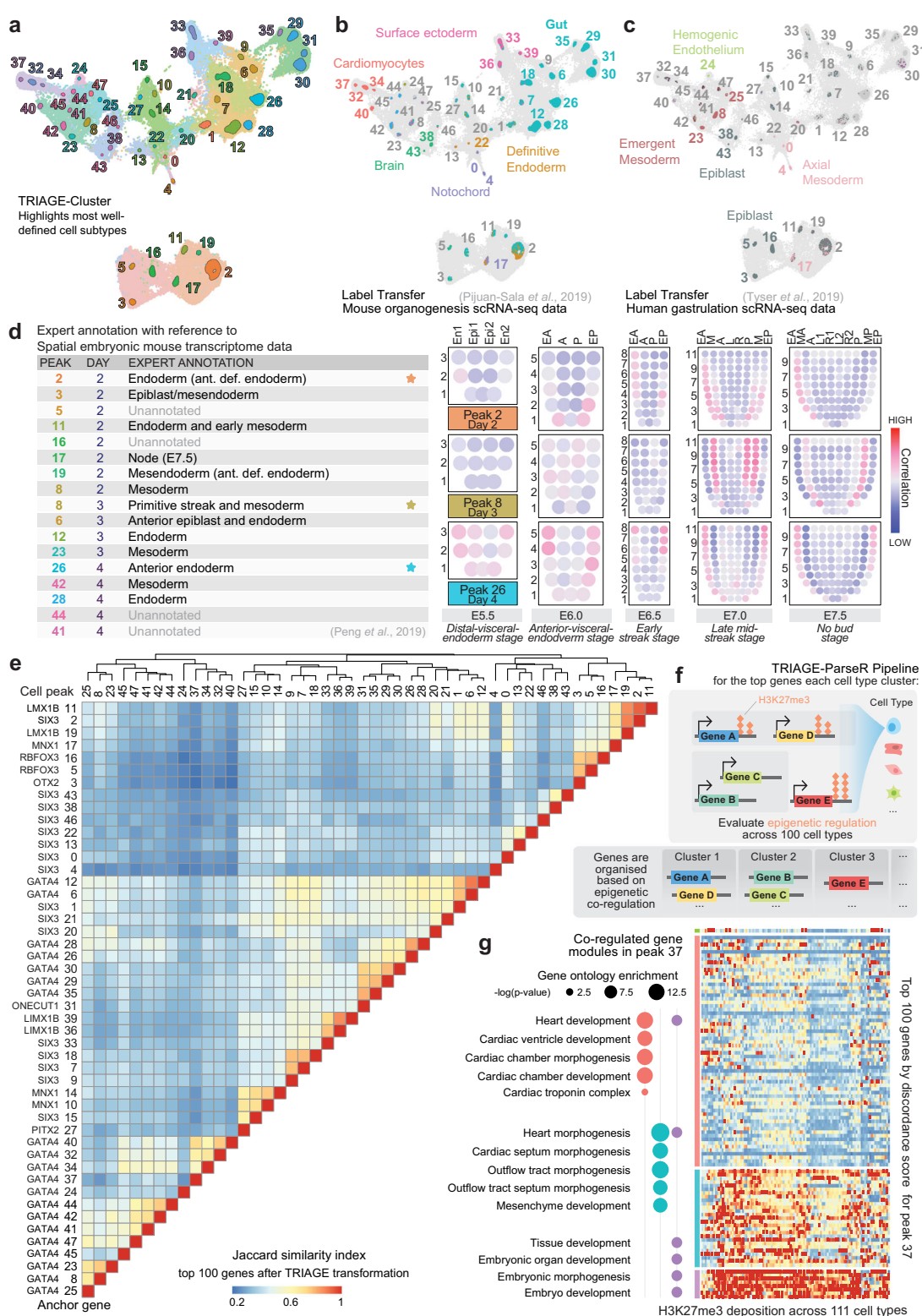

positive gene sets defining seven major cell type domains modelled from 921 RNA-seq datasets[30]. We found that among the high-scoring cells, 5 general cell type domains are represented in our data: mesoderm, surface ectoderm, neural crest, germ, and embryonic domains (Supplementary Fig. 3c). We also used label transfer analysis to score in vitro derived cells based on their transcriptional similarity to single-cell atlases of mouse organogenesis[2] (Fig. 3b) and human gastrulation[4] (Fig. 3c). These reference datasets provide a preliminary indication of the in vitro-derived cell type diversity at the mid- to late timepoints

(days 5-9), revealing a range of high-confidence matches to cell types including cardiomyocytes, hemogenic endothelium, and gut progenitor cells (Fig. 3b, c), aligning with the broad cell type annotations from Fig. 2d. For the earlier timepoints (days 2-4), we assessed transcriptomic similarity of the cell peaks to embryonic mouse spatial transcriptome data[31], revealing early cell types corresponding to gastrulation-stage cell types including those in the epiblast, primitive streak, and mesendoderm (Fig. 3d and Supplementary Fig. 4).

**Fig. 3 | Characterisation and annotation of in vitro-derived cell types. a** Cell types identified from TRIAGE-Cluster compared to broad clusters in Fig. 2d. **b**–**c** Annotation of TRIAGE cell type peaks using in vivo single-cell datasets of (**b**) mouse organogenesis[2], and (**c**) human gastrula[4]. Cells are coloured by their assigned label after thresholding (threshold = 0.5). Dark grey cells lack a label above the threshold. Cell peak labels are coloured by predominant annotation, with grey indicating cell type peaks where most cells score below the threshold. Light grey cells fall outside cell type peaks. See also Supplementary Fig. 3c. **d** Annotation of cell peaks by spatial gene expression domains in mouse embryogenesis from embryonic days (E)5.5 to 7.5[31,83]. Pearson correlation between the expression of selected genes[75] for each TRIAGE peak and embryonic domain is visualised in corn plots. Highly correlated domains for each cell peak and timepoint are summarised in the table (left) and starred rows' corresponding corn plots are displayed (right), where each spot represents expression approximately 5-40 cells. Ant. def. endoderm: Anterior definitive endoderm; En1 and En2: divided endoderm; Epi1 and Epi2: divided epiblast; EA: anterior endoderm; A: anterior; P: posterior; EP: posterior

endoderm; M: whole mesoderm; L: left lateral; R: right lateral; MA: anterior mesoderm; L1: anterior left lateral; R1: anterior right lateral; L2: posterior left lateral; R2: posterior right lateral; MP: posterior mesoderm. See also Supplementary Fig. 4. **e** Pairwise Jaccard similarity score between each TRIAGE-Cluster cell peak using the top 100 genes by TRIAGE-transformation. Anchor genes for each cell peak are listed. **f** TRIAGE-ParseR[32] identifies gene clusters, revealing functional and characteristic gene programmes of cell subtypes. **g** Example TRIAGE-ParseR analysis of cardiomyocytes (peak 37). Heatmap shows H3K27me3 deposition across 111 cell and tissue types in the NIH Epigenome Roadmap across the 100 top genes for cell peak 37 after TRIAGE-transformation. Genes are grouped using the TRIAGE-ParseR and enrichment (-log($p$-value), by one-tailed hypergeometric test, adjusted using Benjamini-Hochberg FDR correction) of select GO terms for each main gene cluster are shown (left). See Supplementary Data 5 for full gene list and Supplementary Data 8 the enriched GO terms for all cell type peaks. Source data are provided as a Source data file.

## Evaluation of gene programmes underpinning cell subtypes

We next used TRIAGE to evaluate cell type-specific gene expression patterns. TRIAGE analysis can return a list of the most cell type-defining genes of any given group of cells, akin to differentially expressed or highly variably expressed genes, which can be used as input for common annotation analyses including gene ontology (GO) or KEGG pathway enrichment. (Supplementary Data 6) Unlike differentially expressed genes however, this approach allows each cell group's expression to be represented without comparison to any other cell cluster in the dataset. We show that the top 100 most highly ranked TRIAGE genes for each cell type can be used to inform regulatory relationships between cell types by evaluating shared and unique genes for each cell type peak (Fig. 3e).

Using these 100 top ranked TRIAGE genes for each cell type, we implemented TRIAGE-ParseR[32], which identifies co-regulated molecular processes driving cell type identity (Fig. 3f). By separating gene lists into related sub-groups, this analysis can enhance enrichment readout from methods such as GO or STRING network analysis and thus clarify cell type annotation (Supplementary Data 6-8). As an example, TRIAGE-ParseR analysis of cell peak 37 reveals three main gene modules with GO enriched terms for cardiac chamber development and morphogenesis of cardiac septa (Fig. 3g), suggesting its transcriptional profile aligns with the embryonic ventricular septum.

Annotation of cell types was performed by drawing on all the provided results to assign cell type annotations to each TRIAGE peak, taking into consideration expression of marker genes, evaluation against in vivo reference points, the top TRIAGE identity genes, and enriched TRIAGE-ParseR gene programmes in each cell type. These results provide an integrated view of iPSC in vitro multilineage differentiation, to facilitate interrogation of temporal and signalling-dependent cell subtype differentiation (Fig. 4a, b and Supplementary Fig. 3d).

The annotated cells in Fig. 4a, b reveal gastrulation stage cell types at day 2 including epiblast, ectoderm, and mesendoderm. Time course data between days 3-5 reveal the emergence of paraxial, cardiac, and forelimb mesoderm together with definitive endoderm and posterior foregut. A sampling of time course and signalling perturbation data on day 5 reveal the first appearance of endocardial endothelium, cells of the anterior foregut, notochord, and neural tube. As cells progress between days 6 and 9 in the time course, we identify the emergence of first and second heart field cardiac lineage cells, and cells of posterior foregut lineages, including pancreas and liver. On day 9, integrated analysis of time course and signalling data show differentiation of additional definitive cell types including ventricular cardiomyocytes, and more developed progenitors of the hepatopancreatic, anterior foregut, and neural lineages. Collectively, these wet and dry lab

methods enable the elucidation of mechanisms controlling multilineage differentiation from pluripotency.

## Analysis of signalling pathway-associated cell differentiation to identify genetic regulators of cell fate

We next assessed differentiation lineage biases arising from each signalling perturbation (Fig. 4c and Supplementary Fig. 5a, see also Fig. 2b). At day 5, cells are broadly distributed across definitive endoderm, foregut endoderm, axial, and general mesoderm. At day 9, however, marked differences emerge. Cells from the control groups show preferential differentiation of posterior foregut-derived hepatopancreatic progenitors, as well as a small proportion of cardiac progenitor cells. This co-differentiation of foregut and cardiac lineages is aligned with embryonic development, where shared signalling cues induce differentiation of the adjacent progenitor populations[33]. Treatments with a lower dosage of XAV393 (low XAV) and recombinant VEGF produce similar proportions of cell subpopulations to the control at day 9, suggesting that these two treatments are at insufficient levels to affect differentiation at that post-gastrulation mesendoderm stage.

In contrast, differentiation of anterior foregut endoderm was observed in conditions exposed to treatment with the WNT agonist CHIR-99021 (CHIR) as well as all BMP signalling perturbations (Fig. 4c). Studies have demonstrated an anteriorising role for BMP inhibition in specifying the foregut endoderm, aligning with the observed effect in the three BMP inhibitory treatments (0.4uM and 4uM Dorsomorphin, and K02288 treatments)[34]. For the anticipated posteriorising WNT and BMP4 signalling cues, it is possible that endogenous feedback mechanisms are activated to correct the effects of the CHIR and BMP4 treatments resulting in similar cell type proportions to the BMP inhibitor treatments at day 9. Lastly, Fig. 4c shows that higher dosage of WNT inhibitor XAV distinctly promoted differentiation of cardiovascular lineage cell types at the expense of the foregut endoderm lineages, in line with established protocols for cardiac-directed differentiation[35].

To leverage this multilineage atlas to discover genetic factors governing cell differentiation, we first compared differentiated cell types from each signalling perturbation at day 9, evaluating pairwise transcriptional correlation and differential gene expression between treatments (Fig. 4d, e). These data demonstrate that modulation of the WNT pathway has the most significant impact on differentiation trajectory, forming the most distinct cell types by day 9. We therefore used differential gene expression between the two WNT-related treatment conditions affecting on day 9 to identify candidate mediators of WNT-requisite cell fate specification, utilising gene ontology annotations to focus on genes known to influence WNT signalling and its effects (Fig. 4f, $y$-axis and Supplementary Fig. 5b). This narrowed the list to a set of candidate genes associated with XAV and CHIR

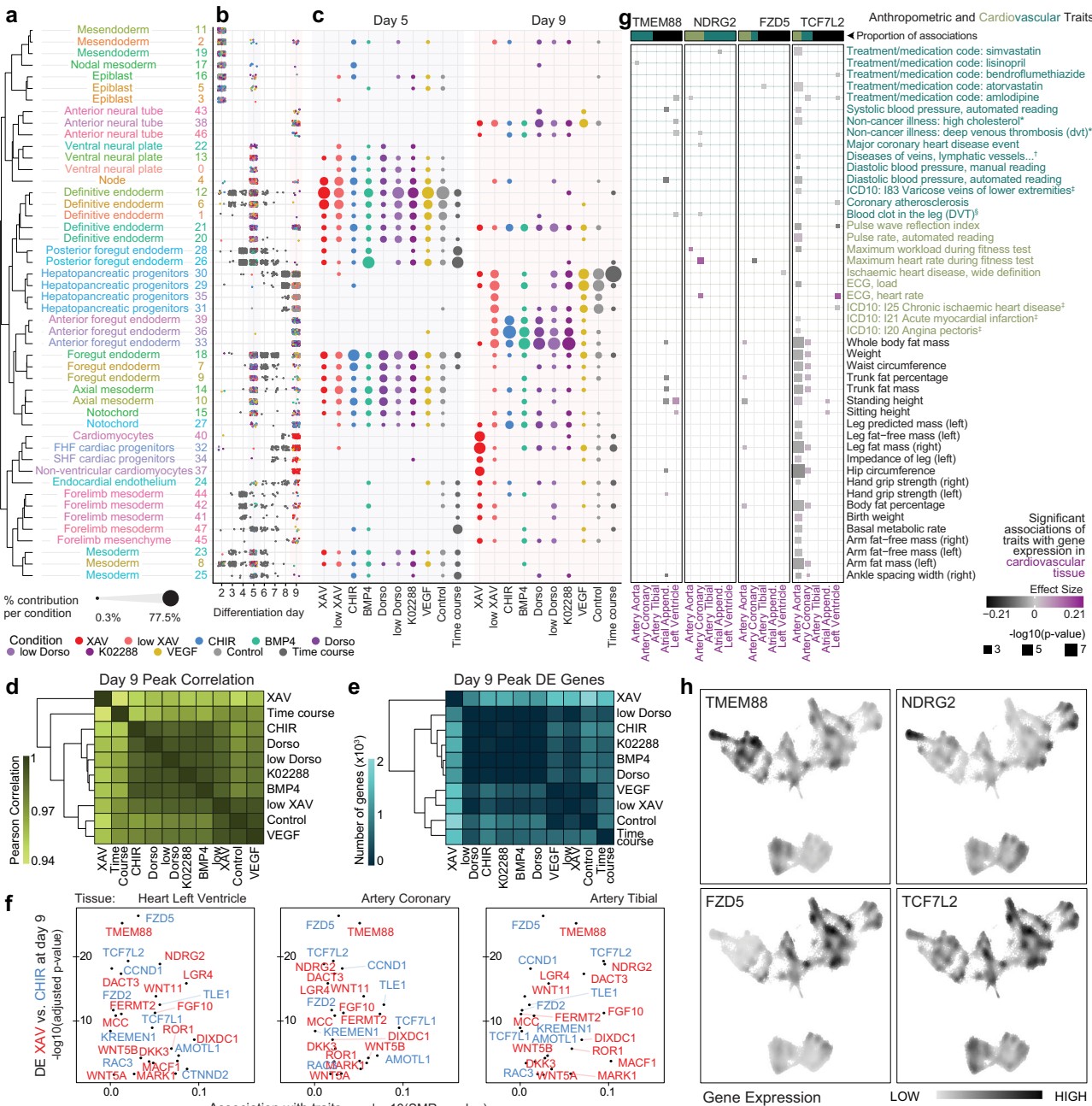

**Fig. 4 | Signalling analysis of cell differentiation. a** Final cell type peak annotations based on Fig. 3b–g workflow. Dendrogram (left) reflects pairwise Jaccard similarity from the top 100 genes after TRIAGE-transformation[29] for each cell peak. Colours correspond to cell peak colours in Fig. 3a. FHF: First heart field; SHF: Second heart field. See also Supplementary Fig. 3d for gene markers supporting each annotation. **b** Every TRIAGE-Cluster peak cell in the atlas across time points, coloured by the signalling condition from which each cell was derived. **c** Proportion of cells from each treatment condition contributing to each cell type peak at days 5 and 9 of differentiation. Each column represents a treatment condition and sums to 100% across all cells produced by that treatment for the corresponding day of differentiation. **d**–**e** Heatmaps summarising the pairwise correlation (**d**) and number of differentially expressed genes (**e**) between cells from each treatment condition at day 9. **f** WNT-related genes that are differentially expressed between XAV (red) and CHIR (blue) treated TRIAGE-Cluster peak cells at day 9 (y-axis) and the significance of the cardiovascular trait most strongly associated with their

expression in three cardiovascular tissues, as determined by Summary data-based Mendelian Randomisation (SMR) analysis (x-axis). DE p-values (y-axis) are determined by Wilcoxon Rank Sum test and corrected for multiple testing using Bonferroni correction with all genes in the dataset. SMR p-values (x-axis) are derived from the SMR test statistic (approximate $\chi^2$) (see "**Methods**"). **g** SMR analysis evaluating associations of candidate WNT-related genes' expression in cardiovascular tissues with cardiovascular and anthropometric traits in the UK Biobank. Some trait and tissue names have been shortened from the following: *Non-cancer illness code, self-reported: …; †Diseases of veins, lymphatic vessels and lymph nodes, not elsewhere classified; ‡Diagnoses – main ICD10: …; §Blood clot, DVT, bronchitis, emphysema, asthma, rhinitis, eczema, allergy diagnosed by doctor: Blood clot in the leg (DVT); Atrial Append.: Atrial Appendage. P-values determined as described for (**f**). **h** Nebulosa[74] plots showing smoothed expression of four candidate WNT-related genes in the iPSC atlas of mesendoderm differentiation. Source data are provided as a Source data file.

modulation of the WNT pathway, primarily mediating cardiac vs. foregut lineage differentiation respectively.

To link these genes to a broader physiological context, we used Summary data-based Mendelian Randomisation (SMR) analysis to evaluate associations between expression of each gene in cardiovascular tissue and effects on genetically-driven anthropometric and cardiovascular traits (Fig. 4f, x-axis)[36]. We selected the most significantly differentially expressed WNT-related genes in the XAV (TMEM88 & NDRG2) and CHIR (FZD5 & TCF7L2) treatments as candidates for further investigation into specific phenotypes identified from the SMR analysis. We found significant associations between eQTL-mediated changes in TMEM88, NDRG2 and FZD5 expression with myocardial and vascular traits including heart rate and deep venous thrombosis (Fig. 4g). The bulk of associations for TCF7L2 stem from variation in its expression in the aorta to affect anthropometric traits including hip circumference and body fat measures, in line with its ties to type 2 diabetes[37]. Given the expression of FZD5 and TCF7L2 in the endodermal foregut lineage cell types (Fig. 4h), we also performed SMR focusing on foregut-related tissue and phenotypes, which showed some enrichment for respiratory phenotypes including asthma (Supplementary Fig. 5c).

Of the four candidate genes, FZD5, TCF7L2, and TMEM88 are also significantly upregulated between days 2 and 5 of differentiation, supporting a more direct role during Wnt-driven fate specification at the progenitor stage (Supplementary Fig. 5d). FZD5 and TCF7L2 have well-established roles in the WNT signalling pathway across foregut, intestinal, and neural development, with the former encoding a receptor in the canonical WNT pathway[38,39] and the latter a transcription factor effector of WNT signalling[40-42]. NDRG2 has been shown to inhibit β-catenin target genes in heart, brain and liver development, as well as being relevant in heart failure contexts[43-45]. While prior work has implicated TMEM88 as a WNT inhibitor relevant in cardiac and pharyngeal pouch development[46-49], the SMR analysis indicates its expression in cardiac tissue being most relevant to vascular, rather than myocardial phenotypes in human populations (Fig. 4g). Furthermore, interrogation of organism-wide gene expression databases shows that TMEM88 expression localises primarily to vascular endothelium in adult organisms (Supplementary Fig. 5e–f), leading us to identify TMEM88 as a candidate with potential for discovery into a broader role in cardiovascular biology.

### Role of *TMEM88* in cardiovascular cell differentiation and physiology

To investigate the role of TMEM88 on differentiation in vitro, we used CRISPRi to generate a doxycycline-inducible TMEM88 knockdown (KD) hiPSC line (Fig. 5a, b). We generated a high resolution scRNA-seq time course of mesendoderm differentiation to evaluate the effect of TMEM88 KD (gain of WNT signalling) compared to control conditions and WNT inhibitory conditions (5 μM XAV) (Fig. 5c, d). We captured 24,841 cells across eight time points between days 2 and 9 of differentiation (Fig. 5d). UMAP plots show marker genes across the time course dataset and conditions (Fig. 5e). Using data analysis workflows described in Fig. 2 and Fig. 3, we identified and annotated 23 cell peaks across the dataset (Fig. 5f, g and Supplementary Fig. 6a–e). Comparing the proportion of cells allocated to each cell type over time for each treatment, we identified notable perturbations in multilineage cell differentiation in TMEM88 KD cells including persistence of lateral plate and paraxial mesoderm as well as depletion of endothelium and posterior foregut and liver bud progenitor cell types (Fig. 5g). Genetic validation studies using antibody-based fluorescence activated cell sorting confirmed that, compared to control iPSCs, TMEM88 KD reduced differentiation of endothelial and cardiac cells in vitro (Fig. 5h and Supplementary Fig. 6f), further supporting a role for TMEM88 in cardiovascular cell identity and differentiation.

To evaluate the role of TMEM88 in cardiovascular development in vivo, we generated a Tmem88 conditional knockout mouse model by crossing a floxed Tmem88 mouse (Fig. 5i) and a mouse with ubiquitous Cre recombinase expression driven from a CMV promoter. Quantitative RT-PCR analysis of heart tissue showed genotype-dependent loss of Tmem88 gene expression (Fig. 5j). All genotypes were born in Mendelian ratios and displayed no observable gross developmental phenotypes (Supplementary Fig. 6g). However, micro-computed tomography scans of E15.5 embryos showed significant changes to organ volumes and structures including the vertebrae, pelvic girdle, and ductus arteriosus in Tmem88 KO embryos (Supplementary Fig. 6h).

Focusing on cardiovascular physiology, analysis by echocardiography in adult mice showed few changes to cardiac structure and function (Fig. 5k and Supplementary Fig. 6i). Two-dimensional b-mode imaging showed significant reduction in left ventricular end systolic volume while pulsed-wave doppler imaging captured a reduction in the E/E' ratio in Tmem88 KO mice (Fig. 5l). Together, these parameters suggest a minor role for Tmem88 in cardiac function. Lastly, based on SMR results that indicated significant associations between TMEM88 and blood pressure phenotypes across diverse tissues (Fig. 5m and Supplementary Data 9), we evaluated systemic arterial blood pressure in WT and KO littermates using catheter-based conductance micro-manometry analysis of pressure traces from the right carotid artery (Fig. 5n). These data show that, compared to WT controls, Tmem88 KO mice had significantly increased variability in systolic and diastolic blood pressure, a phenotype associated with diverse clinical pathologies including hypertension and end-organ damage[50]. Together, these data implicate TMEM88 in developmental regulation of blood pressure, providing genetic evidence of its tissue specificity and functional significance that justify further work to understand the biological mechanism of action. Collectively, this work demonstrates the power of coupling large-scale multilineage iPSC differentiation in vitro with population-level human statistical genetic analysis of complex traits as a basis to discover genetic regulators of developmental physiology.

## Discussion

This study generated barcoded hPSCs to facilitate multiplexed scRNA-seq analysis of cell differentiation, demonstrated by generation of a single-cell transcriptomic atlas chronicling differentiation of pluripotent stem cells in vitro. We further implement a two-part computational approach to cell clustering, including mapping to in vivo cell types and classification of well-defined cell subtypes across diverse lineages to provide, at single-cell level, the molecular attributes of multilineage cell differentiation. We couple these data with human population-level complex trait data to link candidate genetic regulators of development with insights into putative physiological functions in adult mammals. Using genetic models, we validate a role for TMEM88 in governing cardiovascular development, with a primary impact on regulating mammalian blood pressure.

This study provides diverse tools and harnesses data to maximise knowledge-gain into mechanisms of development and disease. First, we develop a cell barcoding strategy. Unlike existing scRNA-seq sample multiplexing strategies[13-15], endogenous cellular barcodes broaden the experimental capability for in vitro and in vivo studies of hPSC differentiation[51]. Barcoding isogenic lines via gene targeting to the AAVS1 locus allows for constitutively expressed barcodes in all cell types for direct comparison of perturbations, reducing variability related to reprogramming and genotype differences between hPSC[52,53] samples and simplifying sample preparation. The CRISPR-Cas9 integration of stable genomic barcodes in this study notably differs from barcoding methods that utilise viral barcode delivery into cells for isogenic multiplexing and lineage tracing applications[51,54]. Lentiviral barcode delivery is highly flexible and allows for successive rounds of barcode incorporation over the course of an experiment to allow direct capture of ancestor-descendant relationships during

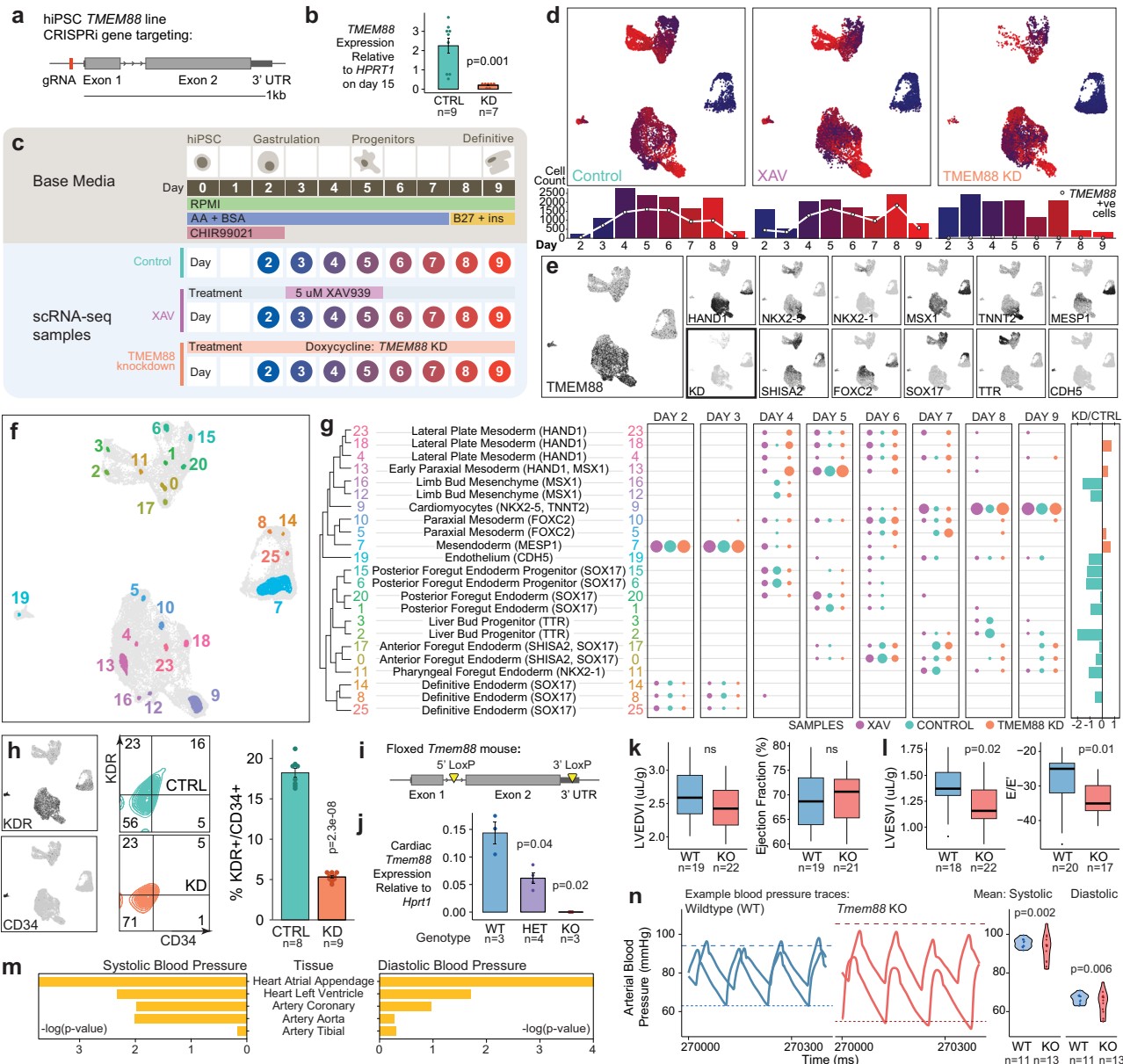

**Fig. 5 | WNT-inhibitor *TMEM88* as a regulator of cardiovascular differentiation and physiology. a–b** Generation of doxycycline-inducible *TMEM88* KD hiPS cell line (**a**) and qPCR confirmation of knockdown efficiency (**b**). **c** scRNA-seq of control, XAV-treated, and *TMEM88* KD differentiation. **d** UMAPs of each sample, cells coloured by collection timepoint. White points on the bar plot (below) show number of *TMEM88*-positive cells. **e** Marker gene expression across all samples including of *TMEM88* (left) compared to knockdown sample (KD, bottom left in bold). See also Supplementary Fig. 6a. **f** TRIAGE-Cluster cell type peaks after filtering (Supplementary Fig. 6b-c). **g** Annotation and relevant marker gene for TRIAGE-Cluster cell types (left) and proportion of cells from each sample over time (right). Dendrogram reflects pairwise Jaccard similarity of the top 100 genes after TRIAGE-transformation[29] for each cell peak. Far right panel quantifies ratio of cells from KD vs. control. **h** Expression of endothelial markers KDR and CD34 in scRNA-seq data (left) and by flow cytometry (middle and right) of control vs *TMEM88* KD at day 5 of differentiation. **i–j** Design for floxed *Tmem88* mice (**i**) and validation of

genotype-dependent expression of *Tmem88* in cardiac tissue (**j**).
**k–l** Echocardiography of 8 week old *Tmem88* KO mice vs wildtype littermates. Left ventricular end diastolic/systolic volume index (LVEDVI/LVESDVI) are normalised to body weight. Box plot centre line shows median, box limits show upper and lower quartiles, whiskers show 1.5x interquartile range, and points show outliers. **m** Summary-data based Mendelian Randomisation significance of *TMEM88*-associated eQTLs from cardiovascular tissues on systolic and diastolic blood pressure. **n** Raw blood pressure traces in left carotid artery by conductance micromanometry and mean systolic and diastolic pressures for adult (aged 8-28 weeks) *Tmem88* KO versus wildtype mice. Dashed line indicates the highest mean systolic blood pressure and dotted line the lowest mean diastolic blood pressure per group. CTRL: control; KD: *TMEM88* knockdown; WT: wildtype; HET: *Tmem88* heterozygote; KO: *Tmem88* knockout. Bar plots are mean ± SEM, *p*-values by two-tailed Welch's *t*-test (**b, h, j**), two-tailed student's *t*-test (**k, l**), SMR test (**m**), and Bartlett test for homogeneity of variances (**n**). Source data are provided as a Source data file.

differentiation and reprogramming[55]. As the genomic barcodes are introduced prior to the start of an experiment, they are limited to lower throughput clonal tracking experiments, or those where separately-prepared and barcoded progenitor cells are pooled for co-culture. This method however has the benefit of a defined barcode integration location, minimising potential for off-target effects and

more consistent expression across cells and cell types. This proves beneficial in contexts involving high cell type heterogeneity or significant changes in cell states, where lentiviral barcodes are more susceptible to cell type-specific silencing. Overall, as pre-experimental sample multiplexing methods, both barcode delivery methods have complementary use-cases and allow scalable, complex experimental

designs for high throughput protocol development, organoid modelling, and in vivo cell transplantation studies.

Cell type annotation remains one of the most challenging issues in single-cell studies due to the need to relate cell heterogeneity to known biology. In this study, we implement a computational analysis pipeline outlined in a package of tools (TRIAGE[29], TRIAGE-Cluster, and TRIAGE-ParseR[32]) to define evolving cell types in vitro by referencing identity-defining genetic features. By restricting the analysis to biologically grounded cell types, we eliminated noise arising from technical variability in scRNA-seq capture and hPSC culture which complicate data interpretation. We demonstrated that this method facilitates more precise predictions of cell identity relationships. Furthermore, while machine learning approaches for cell type annotation are increasingly popular due to their automatic, unsupervised classification of cell types[56], they rely heavily on prior knowledge of cell type annotations. This is problematic particularly among intermediate cell types such as those derived at early stages of in vitro differentiation from pluripotency. This study demonstrates that TRIAGE provides an unsupervised pipeline to evaluate cell types in single-cell data atlases spanning developmental and mature cell types.

Transcriptional analysis of in vitro multilineage cell diversification provides insights into genetic regulation of cell decisions but has limited capabilities for interpreting tissue-level physiological attributes. We demonstrate the value of complementing in vitro atlas data with complex trait genetic data from adult human populations to inform physiological studies of cardiovascular development and disease. In particular we highlight the potential of this approach to facilitate discovery of mechanisms and roles for minimally-characterised genes such as *TMEM88*. While we identify *TMEM88* as a WNT antagonist that regulates cell fate specification, prior studies have shown that WNT inhibition treatment only partially rescues the effects of its loss-of-function in cardiac development, suggesting possible function through other pathways unrelated to WNT signalling in development. Further, given the mismatch of phenotypes between in vitro and in vivo *TMEM88* loss-of-function likely arising from compensatory mechanisms masking more subtle or complex phenotypes in vivo[48,57], reference to independent statistical genetics data provided a novel hypothesis for *TMEM88* in an adult context. Our validation of this role in vivo reinforces the relevance of studying mechanisms that regulate lineage differentiation in broader developmental and disease contexts[58]. This study and approach therefore provide evidence for a novel drug target to address the ongoing health burden of cardiovascular disease.

Blood pressure variability is a significant risk factor in organ damage and a major determinant for diseases including atherosclerosis, chronic kidney disease, and dementia[50,59,60]. Though it is more common among the 33% of adults experiencing hypertension globally[61], blood pressure variability is independently associated with cardiocerebrovascular events including stroke and myocardial infarction, the two leading causes of death worldwide[62,63]. Analysis of one of the world's largest population-based assessments of middle-aged adults with measured blood pressure found that most treated hypertensives remain uncontrolled[64], and only a subset of antihypertensive drugs have demonstrated capability to target both high and variable blood pressure[65]. As there are also no specific treatments for blood pressure variability[66], identification of novel gene targets governing blood pressure, such as *TMEM88*, provides new opportunities to address this persistent public health concern.

## Methods
### Ethics statement
All human pluripotent stem cell studies were carried out in accordance with consent from the University of Queensland's Institutional Human Research Ethics approval (HREC#: 2015001434). All mouse breeding was conducted in accordance with consent from the University of Queensland's Molecular Biosciences Animal Ethics committee

approval (Molecular Biosciences AEC-MBS approval #: 2018/AE000177 & 2021/AE000421). All mouse experiments were carried out in accordance with consent from the University of Queensland's Molecular Biosciences Animal Ethics committee approval (Molecular Biosciences AEC-MBS approval #: 2018/AE000171 & 2021/AE000999).

### hiPSC culture and standard monolayer mesendoderm differentiation.
Undifferentiated hiPSCs were cultured on Vitronectin XF (STEM-CELL Technologies #07180) coated plates in mTeSR1 media with supplement (STEMCELL Technologies #05850) at 37 °C with 5% CO$_2$ and passaged at a ratio between 1:10 and 1:20. On the day prior to differentiation (day -1), cells were dissociated using 0.5 mM EDTA solution and seeded onto separate coated plates in pluripotency media with ROCK Inhibitor (STEMCELL Technologies #72308) and cultured overnight. Once forming an ~80% confluent monolayer, differentiation was induced (day 0) by changing the culture media to RPMI (ThermoFisher #11845119) containing 3 μM CHIR99021 (STEMCELL Technologies, #72054), 500 mg/mL BSA (Sigma #A9418), and 213 mg/mL ascorbic acid (Sigma #A8960). On days 3 and 5, the media was replaced with the same media cocktail excluding the CHIR99021 supplemental cytokine. On day 7 and every subsequent other day, the cultures were fed with RPMI containing 1 × B27 supplement plus insulin (ThermoFisher #17504001).

### Barcode cassette design
The barcodes (Table 1) were introduced into the cells as a part of a barcode cassette, which also incorporated the reverse complement of a partial Chromium Read2 adaptor sequence (truncated slightly to allow oligo length of ≤ 60 bp). This Read2 sequence is used as a PCR handle to generate barcode-containing amplicons compatible with Chromium scRNA library preparation. Additionally, restriction enzyme recognition sequences were added or regenerated to enable easy transfer of the cassette between different vectors. The exact structure of the cassette is as follows: EcoRV site 3' 3 bp – MluI site – partial 10x Read2 adaptor reverse complement−15 bp barcode−MluI complementary sequence. Barcoding cassettes were ordered as complementary single-stranded oligos, which could be annealed and ligated into a digested plasmid backbone.

### Vector design
AAVS1-CAG-hrGFP (Addgene #52344) was used as the plasmid backbone. It contains hrGFP under the control of the CAG promoter, and AAVS1 homology arms to allow precise integration by homologous recombination of the donor sequence into the *AAVS1* locus of the genome when paired with the CRISPR/Cas9 system using a well-described guide RNA[67]. The barcode cassette was introduced between the EcoRV and MluI sites of the plasmid between hrGFP and the polyadenylation site, to enable expression as part of the hrGFP transcriptional unit (Supplementary File 1).

### Generation of barcoding plasmids
AAVS1-CAG-hrGFP (Addgene #52344) was digested by incubation with EcoRV-HF (New England BioLabs; NEB #R3195S) followed by addition of MluI-HF (NEB #R3198S) and further incubation. Successful digestion was confirmed by running a small amount on an agarose gel, and remainder was purified using QIAquick PCR Purification Kit (QIAGEN #28104). Top and bottom strands of barcode oligos were annealed by mixing 1uL each of 100 μM oligo in 1x T4 DNA ligase buffer in a volume of 10 μL, heating to 94 °C for 2 min, then allowing to cool to 25 °C at a rate of 1 °C/s. Annealed oligos were further diluted 1 in 10 with nuclease-free water. Ligation of barcode cassettes to vector was performed by combining 100 ng digested plasmid, 4 μL diluted annealed oligos, and 1 μL T4 DNA ligase (NEB #M0202S; 400U/μL) in 10 μL total volume with 1x T4 DNA ligase buffer. Reaction was incubated at 16 °C for 16 h. Separate reactions were performed for each barcode oligo. 3 μL of ligation reactions were added to 20 μL Stellar competent cells

(Takara #636763) and heat shock transformation performed at 42 °C for 1 min in 1.5 mL tubes. 350 μL SOC media was added for recovery, with shaking at 37 °C for 1 h. 100 μL was spread onto selective ampicillin-containing agar plates, which were incubated overnight at 37 °C. 3 colonies were picked from each plate for screening with colony PCR for expected insert in 10 μL reactions using MangoTaq (Bioline #BIO-21083). Colonies showing successful amplification of insert were grown overnight in 5 mL LB broth containing ampicillin, and plasmid purified using QIAGEN Plasmid Miniprep kit. 100% sequence identity was confirmed across the barcode insert by Sanger sequencing using a universal sequencing primer (5'-ttttggcagagggaaaaaga-3'), performed by the Australian Genome Research Facility (AGRF). 50 mL cultures were grown from glycerol stocks of plasmids with confirmed barcode sequence insert, and plasmid was purified using Nucleobond Xtra Midi Kit (Macherey-Nagel #740410.50) to give endotoxin-free, high concentration stock for transfection.

### Stable generation of WTC BC01-BC18 barcoded iPS cell lines

For gene editing, WTC WT-11 hiPSCs (Gladstone Institute of Cardiovascular Research, UCSF; Karyotype: 46, XY; RRID: CVCL_Y803[68,69]) cells were grown to about 50–80% confluency, dissociated using 1xTrypLE and 100-200 K cells were used for each 10 μl reaction of the Neon Transfection System. The transfection mixture included 0.5 μg of barcoding plasmid DNA, 20 pmol AAVS1-taregting sgRNA (protospacer sequence: GGGGCCACTAGGGACAGGAT, chemically synthesised by Agilent technology) and 20 pmol spCas9 protein (IDT). After electroporation with 1 pulse of 1300 V for 30 ms, cells were seeded in mTeSR1 with ROCK Inhibitor (Y-27632; STEMCELL Technologies #72308) and CloneR (STEMCELL Technologies #05888). Selection was performed with 1 μg/ml puromycin, and purified cell lines were frozen down in CryoStor CS10 Cell Freezing Medium (STEMCELL Technologies #07930) and stored in liquid nitrogen.

### Quality control of cell lines

All cell lines underwent quality testing for correct genetic insertion, selection efficiency, pluripotency, chromosomal abnormalities, and mycoplasma contamination before freezing down, storage, and use in the experiments in this study. Genomic DNA from all cell lines was extracted using QuickExtract DNA Extraction Solution (Epicentre #QE09050). Correct targeting of donor construct at the *AAVS1* locus was confirmed by junction PCR using the following primer pair: AAVS1 F1: 5'-ggttcggcttctggcgtgtgacc-3', AAVS1 R1: 5'-tcaagagtcacccagagacagtgac-3'. The purified PCR product was then Sanger sequenced using a universal sequencing primer (5'-cccatatgtccttccgagtg-3') to validate correct barcode insertion in each cell line. Flow cytometry was performed on live cells for endogenous GFP expression and after labelling for the pluripotency marker SSEA3 (BD Biosciences #562706, 1:20 dilution) and a corresponding isotype control. Karyotyping was carried out as a professional service by Sullivan Nicolaides Pathology. iPSCs were grown in a 25 cm² flask to about 70–80% confluency and sent for analysis, 15 cells were examined per culture. Images of pluripotent cells were taken through a Nikon Eclipse TS100 microscope at 40x magnification with a Nikon ELWD 0.3 T1-SNCP filter.

### Single-cell RNA-sequencing of undifferentiated barcoded iPSCs

All barcoded iPS cell lines were cultured in parallel as described above, dissociated using 0.5 mM EDTA and 600 K cells from each cell line were combined. Prior to this, four cell lines were additionally labelling with different TotalSeq-A cell hashing antibodies[15] according to the manufacturer's protocol. The combined sample was transferred into 2% BSA (Sigma #A9418) in PBS, stained with Propidium Iodide and 500 K viable cells were sorted using a BD Influx™ Cell Sorter (BD Biosciences) with FACSDiva software (BD Biosciences, v9.0).

Single-cell RNA-seq libraries were generated using the 10x Genomics Chromium Single Cell 3' v2 protocol, with minor modifications to

the workflow, outlined by Stoeckius and Smibert to capture the fraction of droplets containing the Hashtag Oligonucleotide (HTO)-derived cDNA ( <180 bp). HTO additive primers and Illumina TruSeq DNA D7xx_s primer (containing i7 index) were ordered from IDT, and used according to the cell hashing protocol. Hashtag libraries were quantified using the Agilent Bioanalyzer. Sequencing was performed using the Illumina Nextseq instrument, with gene expression and HTO libraries pooled on a single flow cell using a ratio of 90:10. Raw sequencing data were processed using the 10x Genomics Cell Ranger pipeline to derive gene expression count matrices. HTO-tagged cells were identified and extracted from the fastq files using the CITE-seq-Count[70] programme with default parameters, to generate a count matrix of cells and their respective HTO expression values. This allowed the pooled hashing cells to be independently deconvoluted. From the 3' gene expression data, barcoded cells were identified by the expression of a barcode from the whitelist, which was included in the transcriptome reference with unique identifiers (e.g., 'BC01').

Transcriptome-based quality control filtering was performed, keeping cells with library sizes between 7500 and 60,000 reads, between 2000 and 8000 distinct features, mitochondrial reads making up no greater than 20% of all reads, and ribosomal reads mapped to between 22 and 45% of total reads. The scds (v1.2.0) R package[71] was also used to remove doublets predicted based on transcriptomic features. For this pilot study, a simplistic demultiplexing method for both the genomic barcodes and the cell hashing multiplexing was used where the highest barcode was taken to be the sample identity for each cell, cells with no counts mapped to any barcode were assigned as 'Negative' and cells with two or more barcodes expressing the same maximum count were determined to be 'Doublets' or 'Multiplets'. The agreement between genomic and hashing barcode assignments used is the sum of the double positives and double negatives divided by the total number of cells. Standard normalisation, scaling, and dimensionality reduction of this dataset was done using the Seurat (v3.0) R package to generate the plots.

### Cell hashed high resolution time course of mesendoderm-directed differentiation

The cell line used to generate the time course and *TMEM88* scRNA-seq datasets were WTC CRISPRi TMEM88-g2.3 GCaMP hiPSCs (Karyotype: 46, XY; RRID: CVCL_VM38; generously provided by M. Mandegar and B. Conklin, Gladstone Institute, UCSF)[72]. In brief, guide RNA targeting the transcriptional start site of *TMEM88* in the human genome was cloned into the pQM-u6g-CNKD doxycycline-inducible construct[72] (Forward oligo: 5'-TTG GAG AGC CGC ATT CCA GGA TTA-3'; Reverse oligo: 5'-AAA CTT ATC CTG GAA TGC GGC TCT-3'). The construct was then transfected into WTC CRISPRi GCaMP hiPSCs using the GeneJuice (Novagen), followed by selection using rounds of replating with 10ug/mL blasticidin treatment. Populations were tested for knockdown efficiency of *TMEM88* (Forward primer 5'-CCT ACT GGT CAC CGG ATT CCT-3', Reverse primer 5'- GAC GCC GAT AAA GGG CTC G-3'), normalised to housekeeping gene *HPRT1* by qPCR (see also Quantitative RT-PCR) following continuous 1ug/mL doxycycline (Sigma #D9891) treatment from day 0 of differentiation.

For the time course experiment, cells were not exposed to doxycycline and treated as transcriptionally wildtype. Mesendoderm-directed differentiation as described above was induced in separate plates for each collection timepoint. Cells were dissociated using 0.5 mM EDTA in 2.5% Trypsin (ThermoFisher #15400054) and neutralised with 50% foetal bovine serum (GE Healthcare Life Sciences #SH30084.03) in DMEM/F12 media (Sigma #11320033). $1 \times 10^6$ cells from each sample were labelled with a different TotalSeq-A cell hashing antibody (BioLegend TotalSeq-A2051-8 anti-human Hashtag 1-8, #394601, #394603, #394605, #394607, #394609, #394611, #394613, #394615) as per the recommended protocol[15] and sorted for viability on a BD Influx Cell Sorter (BD Biosciences) using propidium iodide.

$5 \times 10^5$ live cells per time point were collected and pooled for Chromium Single Cell 3' V3 (10x Genomics) reactions following the manufacturer's protocol, targeting $2 \times 10^4$ cells. Gene expression libraries were sequenced on an Illumina NovaSeq 6000 and the cell hashing libraries were sequenced on an Illumina NextSeq 550Dx. The cellranger pipeline (10x Genomics, v.3.0.2) was used to derive gene expression matrices, and CITE-seq-Count (v4.2.1) was used to count reads for each cell hashing oligonucleotide sequence.

### Multiplexing of signalling perturbations during mesendoderm differentiation

18 barcoded iPS cell lines (WTC BC01-BC18) were cultured in parallel as multiple staggered set-ups of mesendoderm-directed differentiation. Cell lines were divided into 2 batches (BC01-BC09 and BC10-BC18) to allow for the capture of biological duplicates for all 9 conditions. Cells were collected for scRNA-seq on day 2 as a reference prior to any perturbations, small molecule or recombinant protein signalling perturbations were introduced on the day 3 media change and removed on the day 5 media change and resulting cells were collected on days 5 and 9 of differentiation. As such on day 3-5 of differentiation, XAV and low XAV-treated cells were treated with 5uM and 1uM of XAV939 (STEMCELL Technologies #72674) respectively; CHIR treatment used 3uM of CHIR99021 (STEMCELL Technologies #72054); Dorso and low Dorso samples were treated with 4uM and 0.4uM of Dorsomorphin (Sigma-Aldrich #P5499); K02288 samples received 0.5uM of K02288 (Cayman Chemical #16678); BMP4 treatment introduced 10 ng/mL Recombinant Human BMP4 (R&D Systems #RDS314BP010); and VEGF samples were treated with 100 ng/mL Recombinant Human VEGF (R&D Systems #RDSPRD29350). In total, 3 sequencing samples were generated, with each consisting of all 18 cell lines. A combination of 2 different timepoints from the two batches was pooled in to allow for easy detection and removal of potential batch effects during downstream analysis (Table 1).

### Signalling perturbation single-cell library preparation and sequencing

Sample pools were assessed for quality using a hemocytometer with Trypan Blue exclusion. Cell viability ranged from 82–88%; cell concentration was between $1.3 \times 10^6$ and $2.1.9 \times 10^6$ cells/mL. Chromium Single Cell 3' v3 (10x Genomics) reactions were performed for each sample according to the manufacturer's protocol, targeting 20,000 cells per reaction. 11 cycles of cDNA amplification were performed in a C1000 Touch thermocycler with Deep Well Reaction Module (Bio-Rad). After clean-up of full-length amplified cDNA, 10 μL was used for the construction of the gene expression library according to the manufacturer's protocol, with 11 indexing PCR cycles. Additionally, 5 μL of full-length amplified cDNA was used to generate a barcoding library for each sample pool. Briefly, a first round of PCR was performed to specifically amplify cDNA regions containing the barcode cassette and append partial P5 and P7 sequencing adaptors. Each reaction contained 1x KAPA HiFi HotStart ReadyMix and 300 nM each barcode_amp_F (5'-ACT GGA GTT CAG ACG TGT GCT CTT CCG ATC T-3') and barcode_amp_R (5'-CTA CAC GAC GCT CTT CCG ATC T-3') primers in a final volume of 50 μL. A 2-step PCR protocol was performed with annealing/extension at 71 °C for 30 s, for six cycles. After a 1.2X SPRI clean-up to remove primers, a second round of PCR was performed with the entire volume of purified product from PCR1. Each reaction contained 1x KAPA HiFi HotStart ReadyMix, 500 nM SI-PCR primer (5'-AAT GAT ACG GCG ACC ACC GAG ATC TAC ACT CTT TCC CTA CAC GAC GCT C-3'), and 5 μL of a unique i7 indexed R primer from Chromium i7 Multiplex Kit (10x Genomics), in a total volume of 50 μL. PCR was performed as for the SI-PCR protocol in the Chromium gene expression library construction workflow. Eight indexing PCR cycles were performed, for a total of 14 cycles over two rounds of PCR. Final barcoding libraries were purified using 1X SPRI beads, and fragment size and library concentration verified along with

gene expression libraries using a BioAnalyzer DNA High Sensitivity Kit (Agilent). Final gene expression libraries were 62-82 nM, with average size 457-494 bp. Barcoding libraries were 30-38 nM, with average size 358-362 bp.

A single pool was prepared from the three gene expression and three barcoding libraries for sequencing. The samples were pooled equimolar within each library type and combined so that the gene expression libraries together made up 90% of the pool, and the barcoding libraries 10%. Samples were sequenced on an Illumina NovaSeq 6000 (NovaSeq control software v1.6.0/Real Time Analysis v3.4.4) using a NovaSeq 6000 S4 Kit V1 (200 cycles) (Illumina, 20027466) in standalone mode as follows: 28 bp (Read 1), 8 bp (I7 Index), and 91 bp (Read 2). Gene expression count matrices were derived using the standard cell ranger pipeline. (10x Genomics, v3.1.0).

### Demultiplexing, quality control, and visualisation of individual scRNA-seq datasets

Barcoding and cell hashing oligonucleotide (HTO) reads were used to assign sample barcodes to each cell were from the separately amplified and sequenced barcode libraries, and cell barcodes without both transcriptome and sample barcoding reads were removed. For each barcode sequencing library, the 'HTODemux' function in the Seurat (v4.0) R package[73] with the default 0.99 quantile cutoff was used to determine the dominant sample barcode for each cell and annotate negative and doublet cells based on their sample barcode reads alone. Cells negative for HTO or genomic barcode assignments were removed. Three transcriptome-based doublet detection methods in the scds (v1.2.0) R package were used to further assign doublet annotations to each cell, and cells labelled as doublets by at least three methods were removed. Transcriptome-based cell filtering as part of the Seurat pipeline was used to remove low quality cells, retaining cells with library sizes between 10,000 and 100,000 reads, feature counts between 2500 and 10,000, fewer than 20% of reads mapping to mitochondrial genes, fewer than 45% to ribosomal genes for the time course dataset. For the signalling perturbation libraries, removed cells had fewer than 2000 and greater than 7500 detected genes; fewer than 5000 and greater than 50,000 total read counts; or mitochondrial reads accounting for greater than 25% of total reads. Following filtering, sample barcodes were assigned to the remaining cells based on the barcode with the highest expression in each cell. Normalisation and UMAP dimensionality reduction of the data were done following the standard pipeline in the Seurat pipeline. To visualise the expression of marker gene expression in the UMAP plots, we used the R package Nebulosa (v0.99.92)[74] which represents gene expression using kernel density estimation to account for overplotting and noise from expression drop-out.

### Integration of single-cell datasets

Integration of the two scRNA-seq datasets was done using the RPCI method as part of the RISC (v1.0) R package[23], following the suggested pipeline to pre-process raw counts. We used only genes expressed in both datasets to generate 15 gene eigenvectors and using the signalling pathway perturbation dataset as the reference dataset, performed integration to return 50 principal components for further analysis. UMAP dimensionality reduction of the integrated data was also done using the RISC package, using 15 components of the 'PLS' embeddings as recommended for integrated values. Gene expression UMAP plots were generated using the Nebulosa R package. Broad cell type clustering was performed using the Seurat (v3.0) with the resolution parameter set to 0.3.

### Cell-cell interaction analysis

The CellChat[25] R package (v2.1.2) was used to infer, analyse, and visualise cell-cell communication networks. We utilised CellChatDB v2, which contains approximately 3,300 validated molecular interactions.

In total, we have a combination of 10 treatment conditions ("XAV", "lowXAV", "CHIR", "BMP4", "Dorso", "lowDorso", "K02288", "VEGF", "no_treatment", and "no_treatment_tc") and three time points (days 2, 5, and 9), resulted in a total of 30 datasets. We applied the CellChat pipeline to each dataset. Briefly, we created the CellChat object based on the corresponding expression matrix and metadata for each dataset with the 'createCellChat' function. Communication probability was computed with the 'computeCommunProb' function, using the default 'triMean' parameter to approximate a 25% truncated mean. This ensures that the average gene expression is set to zero if less than 25% of cells in a group express the gene. The 'filterCommunication' function was then applied to filter cell-cell communications using the default setting, requiring a minimum of 10 cells in each group to infer communication. The inferred cellular communication networks were extracted at both the ligand-receptor and signalling pathway levels using the 'subsetCommunication' function. Aggregated cell-cell communication networks for each dataset were generated using the 'aggregateNet' function. We used CellChat plot functions to visualise three key signalling pathways - WNT, BMP, and VEGF - if they passed statistical significance, along with the top three pathways with the highest probability of cell-cell communication. Network centrality measures from weighted-directed networks were employed to identify the dominant senders, receivers, mediators, and influencers of intercellular communication, using the 'netAnalysis_computeCentrality' and 'netAnalysis_signalingRole_network' functions. In addition, CellPhoneDB (v5.0.0) along with cellphonedb-data (v5.0), comprising a total of ~3000 interactions, was used to infer cell-cell interactions for each of the 30 datasets (Supplementary File 2). We generated counts and metadata files from the Seurat object to serve as input for CellphoneDB. Following the guidelines provided in the CellPhoneDB Jupyter notebooks (https://github.com/ventolab/CellphoneDB/tree/master/notebooks), we employed the 'simple analysis' mode (method 1) to obtain the mean interaction values for each pair of cell clusters, and employed the 'statistical_analysis' mode (method 2) to evaluate the significance of the interactions present in the dataset. Circle plots, including those shown in Supplementary Fig. 2e, were generated using the 'netVisual_aggregate' function in CellChat with the parameter 'layout=circle'.

### Gene regulatory network analysis using pySCENIC

To assess regulon activities within individual cells, we used the Python-based package pySCENIC (v0.12.1)[26,27]. Unlike traditional methods that focus on individual gene expression, pySCENIC determines rank-based regulon activities by inferring co-expression between transcription factor-target modules and identifying enriched cis-regulatory motifs. The analysis began by identifying co-expression modules comprising transcription factors and their potential target genes. This step was performed using the GRNBoost2 algorithm and applied to the gene expression matrix with reference to the hg38 database within pySCENIC. Following the identification of co-expression modules, we performed cis-regulatory motif enrichment analysis to generate candidate transcription factor regulons using cisTarget with the motif collection database version 10 in pySCENIC. Finally, the activity levels of these candidate regulons were quantified at the single-cell level using AUCell. This approach evaluates whether the regulon gene set is significantly enriched at the top of the gene ranking for each cell, providing a measure of regulon activity across different cellular contexts.

### Lineage inference analysis using URD

Differentiation trajectory reconstruction was using the URD method (v1.1.1)[28] following the instructions outlined on the authors' GitHub (https://github.com/farrellja/URD) and utilising default parameters for pre-processing and normalisation of the raw count matrix. For calculation of the diffusion map to determine transition probabilities between cells, 116 nearest neighbours (knn) and a sigma of 20 were used. Next, pseudotime values were assigned to each cell, ordering them along the developmental process. For this, all cells collected from days 2 and 3 of differentiation were used as root cells and 50 flood simulations determined the visitation structure of the lineage specification graph. The URD tree was constructed using all day 9 cells, annotated by their whole-dataset cluster assignment (Fig. 2d), as tip cells (terminal cell populations), with a p-value threshold of 0.001, 8 bins per pseudotime window, 25 cells per pseudotime bin, and "preference" as the divergence method.

### TRIAGE-Cluster analysis

To assign each cell a repressive tendency score (RTS), we identified the expressed gene with the highest TRIAGE RTS in each cell. The RPCI[23]-generated UMAP coordinates were input into the python scipy.stats (v1.7.1) "gaussian_kde" function, using the RTS vector as the "weights" parameter to estimate a probability density function of the repressive tendency across the data in UMAP space. Using a bandwidth of 0.25, a contour representation of the data was generated, splitting the density function into 10 levels. For each level, the spatial clustering algorithm DBSCAN from the python module sklearn (v0.22) was used to identify all spatially separated groups of cells. TRIAGE "peak" regions were found by identifying the DBSCAN clusters with no smaller clusters within them at higher contour levels. Code is available at https://github.com/palpant-comp/TRIAGE-Cluster.

### Cell type annotation with reference to in vivo datasets

For gene set scoring of cell type domain genes from 921 mouse tissue RNA-seq samples[30], R package biomaRt (v2.46.3) "getLDS" function was used to map the mouse Ensembl gene IDs to human Ensembl gene IDs. For each gene set, the "AddModuleScore" function from the Seurat package was used to assign a gene set score for each cell in each cell type peak. After thresholding scores to be greater than 0, the gene set with the highest score was assigned as the cell type domain annotation for that cell. For label transfer, both our query cell type peaks and the reference in vivo datasets[2,4] were TRIAGE[29] transformed as it has been observed to result in higher prediction confidence in label transfer analysis. The label transfer prediction scores were thresholded only to consider cells with scores over 0.5, and the highest scoring cell type was used to annotate each cell. For both gene set scoring and label transfer annotations, each cell peak's overall annotation is defined as the most common label after thresholding and is assigned as NA if most cells do not have a score greater than the respective thresholds.

**Cell type annotation with reference to spatial domains of embryonic cell types in gastrulating mouse embryo.** To annotate early in vitro cell types from days 2-4 in our dataset with reference to early in vivo cell types, we compared we compared the expression profiles of each TRIAGE cell type peak to spatial mouse embryonic transcriptome data that spans embryonic days 5.5 to 7.5[31]. For days 2-4 in our in vitro data, the top 50 cell type marker genes were determined for each time point using Cepo (v1.12.0), which identifies differentially stable genes between cell types using stability metrics including proportion of zeros and coefficient of variation of a log-transformed normalised expression matrix[75]. Cell type peaks with fewer than 20 cells and genes expressed in less than 5% of any peak were excluded from further analysis. Next, we took the union of all cell type marker genes identified by Cepo and calculated the Pearson correlation of their expression profiles with the spatial mouse transcriptome data. This is used as a proxy to assess how well the expression profile of a cluster recapitulates specific cell types in the gastrulating mouse embryo.

### TRIAGE-ParseR analysis

We calculated the average gene expression value for each TRIAGE-Cluster cell type peak and converted these values to the TRIAGE

discordance score (DS) by multiplication of each gene to its repressive tendency score[29]. We used the top 100 genes ranked by DS from each cell type peak to calculate the pairwise Jaccard similarity index between all peaks. Cell peaks were clustered based on Euclidean distance with complete linkage using the pheatmap (v1.0.12) R package. The top 100 DS genes for each cell peak were then parsed into functionally related modules using TRIAGE-ParseR analysis[32]. In brief, principal component (PC) analysis was performed on H3K27me3 deposition breadth data across all genes and cell types in the NIH Roadmap Epigenomics dataset[76], where each PC represents a biologically meaningful H3K27me3 deposition pattern. The top 67 PCs, which capture over 96% of variance in the data, are used to inform probabilistic clustering of the input genes from each cell peak into co-modulated sets using a Gaussian Mixture Model. The optimal number of clusters is determined by Bayesian Information Criterion, and only clusters with significant ($p < 0.001$, one-tailed hypergeometric test) protein-protein interactions as determined by STRING analysis[77] were kept for further analysis. GO term enrichment analysis was performed using a one-tailed hypergeometric test, with $p$-values adjusted for multiple testing using Benjamini-Hochberg FDR correction. Only GO terms with FDR < 0.01 were included in the final results. Code for TRIAGE-ParseR is available at https://github.com/palpant-comp/TRIAGE-ParseR.

### Differential gene expression, KEGG and GO enrichment analysis

For differential gene expression analysis, we utilised the 'FindMarkerGenes' function from the Seurat R package. Heatmap visualisation displaying number of differentially expressed genes between treatments was generated using the pheatmap R package. WNT-related GO term associations were identified using the biomaRt R package, and DE genes were annotated by their most specific directional GO term. GO term and KEGG pathway enrichment analysis for the differentially expressed genes compared between the XAV and CHIR treatments at day 9 was performed using the 'enrichGO' and 'enrichKEGG' functions in the clusterProfiler R package. The 'simplify' function from the clusterProfiler (v4.6.2) R package was used to reduce redundancy of enriched GO terms. WNT-related genes in Fig. 4f and Supplementary Fig. 4d are defined as those associated with the following GO terms: Wnt signalling pathway (GO:0016055), Positive regulation of Wnt signalling pathway (GO:0030177), Negative regulation of Wnt signalling pathway (GO:0030178), Positive regulation of canonical Wnt signalling pathway (GO:0090263), Negative regulation of canonical Wnt signalling pathway (GO:0090090), Positive regulation of non-canonical Wnt signalling pathway (GO:2000052), and Negative regulation of non-canonical Wnt signalling pathway (GO:2000051).

### Generation of *TMEM88* KD scRNA-seq dataset

The WTC CRISPRi TMEM88-g2.3 hiPSCs in each sample were cultured in 8 temporally staggered setups of mesendoderm-directed differentiation. The XAV group was treated with 5uM XAV393 (STEMCELL Technologies #72674) for 48 h, from day 3 to day 5 of differentiation, while the *TMEM88* KD group was treated with 1ug/mL doxycycline (Sigma #D9891) for the entire duration of differentiation starting from day 0. Cells were dissociated using 0.5% EDTA in 2.5% Trypsin (ThermoFisher, #15400054) and neutralised with 50% foetal bovine serum (GE Healthcare Life Sciences, #SH30084.03) in DMEM/F12 media (Sigma #11320033). $1 \times 10^6$ cells from each sample were labelled with a different TotalSeq-A cell hashing antibody (BioLegend TotalSeq-A2051-8 anti-human Hashtag 1-8, #394601, #394603, #394605, #394607, #394609, #394611, #394613, #394615) as per the recommended protocol[15] and sorted for viability on a BD Influx Cell Sorter (BD Biosciences) using propidium iodide. $5 \times 10^5$ live cells per time point were collected and pooled for Chromium Single Cell 3' V3 (10x Genomics) reactions following the manufacturer's protocol, targeting $2 \times 10^4$ cells. Gene expression libraries were sequenced on an Illumina

NovaSeq 6000 and the cell hashing libraries were sequenced on an Illumina NextSeq 550Dx.

**Quality control processing of *TMEM88* scRNA-seq dataset.** This dataset was pre-processed in the same manner as the time course dataset; the cellranger pipeline was used to process raw reads, cell hashing barcodes (time points) were demultiplexed with the 'HTO-Demux' function from the Seurat (v3.0) package, and doublet detection was performed with the scds package, removing cell barcodes described as doublets by at least 3 methods. Transcriptome-based cell filtering as part of the Seurat pipeline removed cells with fewer than 1000 and greater than 10,000 total read counts; fewer than 100 and greater than 300 detected genes; mitochondrial reads comprising greater than 15% of the library; or ribosomal reads making up greater than 45% of the library. Following filtering, time points were assigned to the remaining cells based on their cell hashing barcode with the highest expression in each cell.

### Quantitative RT-PCR

Cells were lysed using β-mercaptoethanol in RLT buffer (QIAGEN #74106) and snap frozen on dry ice. Total RNA was collected with the RNeasy Mini Kit (QIAGEN #74106) according to manufacturer's instructions. RNA was reverse transcribed into cDNA using Superscript III Reverse Transcriptase (ThermoFisher Scientific #18080051). qRT-PCR was performed using SYBR Green PCR Master Mix reagents (ThermoFisher #4312704) on a ViiA 7 Real-Time PCR System (Applied Biosystems). Gene expression was analysed in R (v4.0.4) where relative gene expression levels were calculated for each gene by raising two to the power of the measured gene (averaged over two technical replicates), divided by the mean control housekeeping gene expression. For human samples, *HPRT1* was used as the control (Forward primer 5'-TGA CAC TGG CAA AAC AAT GCA-3', Reverse primer 5'-GGT CCT TTT CAC CAG CAA GCT-3'), and *Hprt1* was used for mouse samples (Forward primer 5'-TCA GTC AAC GGG GGA CAT AAA-3', Reverse primer 5'-GGG GCT GTA CTG CTT AAC CAG-3'). Data were analysed in R, where outlier values (those exceeding mean ± 2 standard deviations) in each group were removed. The statistical test used to compare the normalised expression means from separate experiments was a two-tailed $t$-test with a 95% confidence interval using R.

**Flow cytometry.** Cells were dissociated with 0.5 mM % EDTA + 0.25% Trypsin (ThermoFisher, #15400054) then neutralised with 1:1 foetal bovine serum (FBS, GE Healthcare Life Sciences #SH30084.03) in DMEM/F12 (Sigma #11320033) or RPMI media (ThermoFisher #11845119). Cells were fixed using 4% paraformaldehyde, permeabilised in 0.75% saponin with 5% FBS in PBS and stained for flow cytometry using cardiac troponin T (cTnT, Creative Biolabs #M40283, 1:200 dilution). For live cell cytometry, cells were stained in DMEM/F12 or RPMI media for SSEA3 (BD Biosciences #562706, 1:20 dilution), KDR (R&D Systems #FAB357P, 1:10 dilution), or CD34 (BD Biosciences #340430, 1:5 dilution) with the corresponding immunoglobin G isotype control. Flow cytometry data was captured using a BD FACS-CANTOII with the FACSDiva (v9.0) software. Analysis of this data was performed using FlowJo (v10.07). Summaries of the cell populations were analysed and visualised in R, where outlier values (exceeding mean ± 2 standard deviations) in each group were removed before performing a two-tailed $t$-test for the comparison of means with a 95% confidence interval.

### Summary-data based mendelian randomisation (SMR) analysis

We used 61 anthropometric and cardiovascular phenotypes[36] from the UK Biobank[78], whose heritability estimates were given as "medium" or "high" by Neale's lab, to explore the broad effect of select genes on human traits. Publicly available summary statistics generated by Neale's lab were used to perform Summary-data Mendelian

Randomisation[79] (SMR) analysis using eQTL (expression quantitative trail loci) information derived in the V8 GTEX release[80]. Individuals with European ancestry from the 1000 genomes project were used as a LD (linkage disequilibrium) reference. SMR analysis was performed with a $p$-value threshold for eQTL inclusion of 0.05 to ensure all eQTLs affecting gene expression were captured. SMR $p$-values are derived from the SMR test statistic which is an approximate $\chi^2$ statistic estimated from z statistics from the summary statistics and eQTL data. SMR analysis across all phenotypes was performed on the vascular (coronary artery, tibial artery, aortic artery), cardiac (left ventricle, atrial appendage), respiratory (lung), and gut tube derivative (gastro-esophageal junction, oesophageal mucosa, pancreas, stomach, and thyroid) tissues. Further analysis for *TMEM88* was performed on systolic blood pressure, automated reading (4080) and diastolic blood pressure, automated reading (4079) phenotypes using all 49 GTEX tissues.

**Animals.** All animals used in this study were littermates from the *Tmem88* KO mouse line (described below). All mice were maintained in IVC caging in clean conventional conditions, housed in cages with no more than five mice, and kept in a regular 12 h light/ 12 h dark cycle (lights on at 6:00am). The temperature was $24 \pm 2\,^{\circ}\text{C}$ and humidity was 40–70%. Cages were monitored daily with fortnightly cage change. For mice with heart tissue collected for downstream assays, euthanasia was achieved via thoracotomy and heart excision after deep surgical anaesthesia from intraperitoneal injection of 80–400 mg/kg ketamine + 10 mg/kg xylazine. All other mice were euthanised by cervical dislocation under anaesthesia ($CO_2$ or isofluorane). For all experiments, sexes were pooled in the study design.

### Generation of *Tmem88* conditional knockout mouse line

The floxed *Tmem88* mice were generated by the Monash Genome Modification Platform, Monash University. In brief, CRISPR RNAs were ordered from Integrated DNA Technologies and annealed with trans-activating CRISPR RNA (Alt-R CRISPR-Cas9 tracrRNA from IDT #107319 to form single guide RNA (gRNA) targeting Exon 2 of *Tmem88* (gRNA1: 5′-AAT GGG CCA CTC TGC CCA GG-3′, gRNA2: 5′-TTT GCC CAT TTG TAG TCT TG-3′). 10 ng/uL Cas9 nuclease (IDT Alt-R® S.p. Cas9 nuclease) was incubated with 10 ng/uL gRNAs to form a ribonucleoprotein (RNP) complex. A 1247 bp ssDNA repair template containing LoxP sites flanking the second exon of *Tmem88* (see also Fig. 5i) was generated using Invitrogen Dynabeads™ MyOne™ Streptavidin C1 according to manufacturer's instructions. The RNP and ssDNA repair template (10 ng/uL) were microinjected into the pronucleus of C57BL/6 J zygotes at the pronuclei stage. Microinjected zygotes were transferred into the uterus of pseudo pregnant F1 females.

Floxed *Tmem88* animals were crossed with C57BL/6 mice expressing Cre recombinase driven by a CMV promoter, to generate animals with ubiquitous loss of the *Tmem88* gene. Both hetero- and homozygous mutants were viable and showed no obvious phenotype. For genotyping the animals, DNA was extracted from ear or toe samples using 50uL of QuickExtract (Biosearch Technologies, #QE09050), incubated at 65 °C for 15 min, then 100 °C for 5 mins and diluted to 50 ng/uL for genotyping. Each mouse was tested for genomic presence of Cre-Recombinase (Forward primer 5′-CTG ACC GTA CAC CAA AAT TTG CCT G-3′, Reverse primer 5′-GAT AAT CGC GAA CAT CTT CAG GTT C-3′; Cre = 200 bp), Floxed *Tmem88* (Forward primer 5′-CGG CTC AGA ATC TGC TGG TG-3′, Reverse Primer 5′-TTT CTA GCG GGA AGC GGT GT-3′; Wt = 1300 bp, Flox = 1350 bp, KO = 600 bp), and both LoxP sites (5′LoxP Forward primer 5′-CGG CTC AGA ATC TGC TGG TG -3′, 5′LoxP Reverse primer 5′-GCA GAT GCG AGG TGC AAA GG-3′, 3′LoxP Forward primer 5′-TCC AAT CGC TCC TGC ACT GG-3′, 3′LoxP Reverse primer 5′-TTT CTA GCG GGA AGC GGT GT-3′; Wt = 300 bp, LoxP = 350 bp). Quantitative RT-PCR was also used to confirm the loss of *Tmem88* expression in the mutants (Forward primer 5′-CTG GTG GCT GTT TTC AAT CTC C-3′, Reverse primer 5′-GTG

CCT GAG AGC GCA GAA A-3′). For RNA extraction, 30 mg heart tissue was homogenised in liquid nitrogen using a mortar and pestle before proceeding with the RNeasy Mini Kit (QIAGEN #74106) following the manufacturer's instructions. cDNA synthesis and quantitative RT-PCR were performed as described above.

**Mouse echocardiography.** Blinded echocardiography was performed on 8 week old mice using a Vevo 3100 Preclinial Imaging System (VisualSonics, Toronto, Canada) with a 25-55 MHz linear transducer (MX550D). Mice were anaesthetised using 4% isofluorane, and general anaesthesia was maintained with 1% isofluorane during the procedure. Mice were weighed and placed in supine position on a heating pad during echocardiography with body temperature, heart rate, and electro-cardiogram recorded throughout using Vevo 3100 (v3.8.2) software. Warmed echo transmission gel was applied to the hairless chest of the mouse before collecting images. Two-dimensional B-mode images recorded in the parasternal long axis view were used to determine left ventricular end-diastolic/systolic volume (LVEDV/LVESV) and fractional shortening. One-dimensional M-mode images in the parasternal short-axis view were also used to measure ejection fraction (EF) and thick-nesses of the posterior walls of the left ventricle in diastole (LVPWd). Pulsed-wave doppler recordings in apical four-chamber view were used to measure mitral inflow velocity of blood during passive filling of the ventricle (E-wave) and active filling during atrial systole (A-wave). Pulsed-wave tissue doppler in apical four-chamber view was used to capture mitral annular velocity during early (E′) and late (A′) diastole. Analysis of echocardiography images was performed in VevoLab (version 3.1.1) software (VisualSonics, Toronto, Canada). LVEDV, LVESV, and LVPWd were normalised to body weight to achieve left ventricle end diastolic index (LVEDI/LVESI), and LVPWd index, respectively. For each animal, all parameters were measured on average 4 times (minimum twice) and averages are presented. Animals with abnormal body weights and heart rates were excluded (4 animals removed), all measurements from images with consistent outlier measurements indicating incorrect image capture were excluded (12 images removed), as well as values more than two standard deviations from the mean in each genotype group for each measured parameter (annotated in Source Data). Statistical analysis was carried out using R with the ggpubr (v0.2.3) package to perform a two-sided $t$-test with a 95% confidence interval.

### Mouse blood pressure measurements

Adult mice between 8 and 28 weeks of age were anaesthetised using 4% isofluorane before endotracheal intubation and mechanical ventilation. Isofluorane was reduced to 1.5% to maintain general anaesthesia for the remainder of the procedure. A Millar Pressure-Volume (PV) catheter (Millar #SPR-839, 1.4 F Catheter) was used to cannulate the right carotid artery, to allow a stable pressure trace as measured by the MVPS Ultra Pressure-Volume Unit (Millar, #880-0168). The continuous baseline blood pressure was measured for 5-10 min, and the average maximum (systolic) and minimum (diastolic) pressure was recorded using the LabChart Pro v8 software (ADInstruments, Dunedin, New Zealand) twice for each animal. Animals with any average measurements more than two standard deviations away from the respective systolic or dia-stolic genotype mean were excluded (2 animals removed).

### Micro-computed tomography and automated image analysis (LAMA)

Embryos were dissected at embryonic day (E)15.5 in PBS, rinsed in cold water and stained in 0.7% phosphotungstic acid (Sigma Aldich #P4006) diluted in 70% ethanol at RT for 14 days[81]. Stained embryos were mounted in agarose and scanned on a Skyscan 1272 micro-CT scanner (Bruker) using an Al 0.5 + Cu 0.038 filter and voltage of 87 kV. Images were captured at 1640 ×;2452 resolution (6 μm³ voxel size) through 360° at 0.4° angle increments and reconstructed with smoothing, ring artefact reduction and beam hardening options in

NRecon software (Bruker). 3D datasets were downsampled to 14 μm³ and each registered using the LAMA phenotyping pipeline[82] to an E15.5 population average so that its 43-organ atlas (in preparation) maps onto each embryo dataset. The resulting organ and whole embryo volumes (WEV) were fitted to a linear model organ volume/WEV~genotype + WEV with heterozygous and homozygous *Tmem88* embryos statistically compared to wildtype littermate controls. *P*-values were corrected for multiple comparisons using the LAMA permutation statistics script.

## Statistics & Reproducibility

For hiPSC-based experiments, no statistical method was used to pre-determine sample size. Power analysis was used for animal experiments to determine number of animals required per group to detect a 30–40% difference in gene expression, structural and functional measurements (SD = 0.2, alpha= 0.05, confidence= 95%). Any cases of data exclusion have been outlined in their respective methods section (typically measurements more than two standard deviations from the group mean were excluded). Mouse phenotyping experiments and outcome assessment were carried out blinded. All references to sample size (n) refer to measurements from distinct samples (separate differentiation runs for hiPSC experiments and separate animals for mouse experiments).

## Reporting summary

Further information on research design is available in the Nature Portfolio Reporting Summary linked to this article.

## Data availability

All raw and processed single-cell RNA sequencing data generated in this study have been deposited in the NCBI Gene Expression Omnibus repository (GSE279710). An interactive version of the combined, annotated dataset is also available at http://cellfateexplorer.d24h.hk/, where additional *TRIAGE-ParseR* analysis outputs, metadata, and gene expression for individual cell type peaks can be queried. Published datasets used as references include mouse organogenesis data (E-MTAB-6967); human gastrulation data (E-MTAB-9388); and mouse gastrula development data (GSE186069). Source data are provided with this paper.

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

## Acknowledgements

We thank Nick Valmas for illustrations; the Australian Genome Research Facility (AGRF) for Sanger sequencing; the Flow Cytometry Facility at the Queensland Brain Institute for facilitating cell sorting; the University of Queensland sequencing facility at the Institute for Molecular Bioscience for performing single-cell capture and library preparation; and the Garvan Sequencing Platform for sequencing. The mutant floxed *Tmem88* mice were produced via CRISPR genome editing by the Monash Genome Modification Platform, Monash University as a node of Phenomics Australia (formerly Australian Phenomics Network). Phenomics Australia is supported by the Australian Government Department of Education through the National Collaborative Research Infrastructure Strategy, the Super Science Initiative, and the Collaborate Research Infrastructure Scheme. We acknowledge the Victor Chang Research Institute Innovation Centre, funded by the New South Wales Government. Funding support was provided from the National Health and Medical Research Council of Australia (Grants 1143163 (N.J.P., J.E.P., and S.W.L.), 1110751 (P.P.L.T.), 1173469 (P.Y.), GNT2008928 (Q.N.), APP1107599 (J.E.P.), 2007896 (S.L.D)), the Medical Research Future Fund (APP2016033, N.J.P. and M.B.), the Australian Research Council (190102793 (P.P.L.T.)), the National Heart Foundation of Australia (Grants 101889 and 106721, N.J.P.), and AIR@InnoHK administered by Innovation and technology Commission (J.W.K.H).

## Author contributions

To generate the barcoded iPS cell lines, S.L. designed the barcodes; S.A. and D.X. designed the plasmids. S.A. cloned the plasmids; T.W. and H.C. carried out the cell culture; D.X. designed the CRISPR targeting strategy. T.W. and D.X. performed the CRISPR gene editing. M.T. and D.X. performed the genotyping and Sanger sequencing. For the pilot barcoding scRNA-seq, T.W. and H.C. carried out the cell culture and collection; S.L. and S.A. processed and sequenced the samples; S.L. and S.S. performed the computational sample demultiplexing and quality control analysis. J.E.P. supervised the barcoding design and assisted with sequencing of barcoded libraries and facilitated data interpretation. For the time course and *TMEM88* KD scRNA-seq datasets, H.C. generated the CRISPRi hiPSC line; S.S. and X.C. carried out cell culture, collection, and sample preparation; S.S. performed quality control and preliminary computational analysis. For the signalling perturbation scRNA-seq dataset, T.W. and H.C. carried out cell culture, collection, and sample preparation; S.S. and T.W. completed quality control and preliminary computational analysis. For analysis of the combined dataset, S.S. integrated the data and led analysis; E.S., Q.Z., and Y.S. performed additional characterisation analysis of broad cell clusters; Y.S. and S.S. implemented the TRIAGE-Cluster approach; D.K., P.Y., and P.P.L.T. performed annotation of cell types with reference to spatial gene expression patterns in the developing mouse embryo; D.P. constructed the online dataset visualisation dashboard and website; Z.S. implemented the visualisation of gene expression to the platform; Q.N. supervised creation of the website; J.W.K.H supervised implementation of the gene expression visualisation; W.S. conceptualised and implemented the TRIAGE-ParseR analysis pipeline; M.B. supervised development of the TRIAGE-ParseR software. For analysis of the role of *TMEM88*, D.M. performed statistical genetics analysis of *TMEM88* on human complex trait data; S.N. provided preliminary analysis of GWAS data linking variants in *TMEM88* with vascular physiology; S.K. performed analysis of *TMEM88* KD cardiac and endothelial iPS-derived cell types; E.M.M.A.M. performed micro-computed tomography and data quality control; G.C. performed LAMA analysis micro-computed tomography datasets and interpreted the results; S.L.D. supervised micro-computed tomography and LAMA analysis of the *Tmem88* embryos; S.S. performed echocardiographic imaging and associated analysis with supervision from M.A.R. and J.O.; J.O. performed conductance micromanometry analysis for analysis of blood pressure. N.J.P. conceptualised the study, supervised the project, and raised funding. S.S. and N.J.P. wrote the manuscript.

## Competing interests

The authors declare no competing interests.

## Additional information

[1]Institute for Molecular Bioscience, The University of Queensland, St Lucia, QLD, Australia. [2]Genome Innovation Hub, The University of Queensland, St Lucia, QLD, Australia. [3]Victor Chang Cardiac Research Institute, Darlinghurst, NSW, Australia. [4]School of Clinical Medicine, Faculty of Medicine and Health, University of New South Wales, Sydney, NSW, Australia. [5]School of Biomedical Sciences, Li Ka Shing Faculty of Medicine, The University of Hong Kong, Pokfulam, Hong Kong SAR, China. [6]Laboratory of Data Discovery for Health Limited (D24H), Hong Kong Science Park, Hong Kong SAR, China. [7]Computational Systems Biology Group, Children's Medical Research Institute, University of Sydney, Westmead, NSW, Australia. [8]Charles Perkins Centre, School of Mathematics and Statistics, University of Sydney, Camperdown, NSW, Australia. [9]Queensland Facility for Advanced Genome Editing, The University of Queensland, St Lucia, QLD, Australia. [10]Garvan-Weizmann Centre for Cellular Genomics, Garvan Institute of Medical Research, Darlinghurst, NSW, Australia. [11]University of New South Wales, Cellular Genomics Futures Institute, Sydney, NSW, Australia. [12]Faculty of Science, University of New South Wales, Sydney, NSW, Australia. [13]Embryology Research Unit, Children's Medical Research Institute, University of Sydney, Westmead, NSW, Australia. [14]School of Medical Sciences, Faculty of Medicine and Health, University of Sydney, Camperdown, NSW, Australia. [15]School of Chemistry and Molecular Biosciences, The University of Queensland, St Lucia, QLD, Australia. ✉e-mail: n.palpant@uq.edu.au

