## [Peer review File · Nature Communications]

REVIEWER COMMENTS

Reviewer #1 (Remarks to the Author):

The authors have generated barcoded iPSC lines and performed a WNT-induced differentiation experiment and collected scRNAseq data at 8 time points. They have additionally perturbed WNT, BMP, VEGF signaling along the differentiation trajectory and obtained scRNAseq data at 3 time points. Utilizing their in-house TRIAGE tools, the authors have then annotated the cell types in their atlas and determined the gene programs underlying the annotated cell types. Next, to determine the association between genes and physiological traits, the authors have used Quantitative Trait Loci (QTL) to link their signaling perturbation analyses to physiology. From these analyses, they have identified and further studied the role of TMEM88 in cardiac physiology and blood pressure, showing that TMEM88 KO mice have increased variability in systolic and diastolic blood pressure.

Overall, this study generated a lot of high-quality single cell data and analysis, and this differentiation atlas will be a valuable addition to the field. The analyses and techniques used in this study (Millar Pressure-Volume catheter for instance) are reliable methods in the field. Most importantly, given that blood pressure variability is associated with leading causes of death like strokes and myocardial infarctions, the involvement of TMEM88 in regulation of blood pressure is a fascinating finding for both the advancement of knowledge in the field and in the clinical setting, and merits publication. However, a few concerns would need to be addressed prior to that.

Concerns and suggestions:

1. Figure 4d -- FZD5 and TCF7L2 seem similarly important as TMEM88. Could the authors elaborate why TMEM88 was selected? A clearer metric to determine TMEM88 as a top factor, or testing a few more of the top candidates using the SMR analysis and expression in the atlas will be helpful here.
2. Could the authors provide a graph of the number of cells for each of the treatment conditions at the 3 time points (Figure S1d) (Similar to the graph in S1C top left). This will be essential in evaluating the data presented in Figure 4a where %contribution per condition is used as a metric. It may be that considerably fewer cells are captured in some of the treatment conditions (especially at d9), which would bias the inferences.
3. Figure 4a – Could the authors allude to the reasons why low levels of Wnt inhibition seems to favor the pancreas, liver, tracheal progenitor lineages?
4. Also, in Figure 4a, both the inhibition and promotion of BMP signaling seems to contribute to the same cell types. Could the authors explain why this could be the case? Additionally, I am curious if this could be because using a Wnt-activation based differentiation protocol would overshadow the perturbations from other signaling pathways?

5. Figure 3h works great to showcase the data, and it's remarkable to see the lineage specification in the cells captured as the differentiation timepoints pass. However, page 7, line 161-175 could use more details and elaboration before an integrated view can be presented (line 77). Especially for readers unfamiliar with TRIAGE tools, some more details or stepwise logic of Figure 3e-g will be helpful.

6. If TMEM88 KD reduced differentiation of cardiac cells in vitro, it seems surprising that the TMEM88 KO mice (especially homozygous) show no impact on cardiac structure and function (line 238). Could the authors elaborate on possible reasons for this?

Minor:

1. Table S1 seems crucial to understand the barcoding strategy and experimental outline of the perturbation. My recommendation would be to include it as a panel in Figure 2, or as a main table. This would also avoid the confusion whether each cell line has 1 barcode or 1 full barcode library (containing 10k barcodes) (line 95). Another option could be to move the panel 2b to supplementary or combine it with the Table S1.

2. Figure 1a – The barcoding vs cell vs cost diagram is a bit confusing. Barcoding at single cell level can also include population level barcodes in the cassette, and could also be done at the start of the experiment. Could the authors elaborate what the intended message was in this figure 1a panel?

3. Figure 2a – “Barcode cell editing by CRISPR/Cas9” – should this more aptly say: “Barcode insertion into the cell lines by CRISPR/Cas9” to avoid confusion since the barcodes aren't being edited?

4. Figure 4b-c legend: Should be (b) and (c) instead of the (c) and (d)

Reviewer #2 (Remarks to the Author):

In this study, Shen et. al constructed genetically barcoded iPSCs for sample multiplexing in single-cell RNA-seq experiments and used them to study the differentiation of iPSCs upon standard and multiple drug treatment conditions. Extensive computational analyses were applied to map the identity of differentiated cells, for characterizing the genetic programs driving different lineages and the outcome of each perturbation. With the platform, the authors identified TMEM88 as a putative regulator affecting cardiovascular cell differentiation and related that to cardiovascular and anthropometric traits using gene-traits association analysis, and explored the physiological role of TMEM88 in regulating blood pressure in vivo using KO mice. Taken together, the study represents a

good case study of how one can couple iPSC differentiation models, single-cell RNA-seq, and computation tools to pinpoint novel gene targets of human diseases and cell types affected, with the potential to link genes, cell types, and human diseases in a broad context. The following are several comments that should be fixed before the publication of the manuscript:

1. More technical justifications are required for evaluating the performance of single-cell capture of transcribed barcodes. For example, in line 100, the author mentioned that only 42.0% of cells containing reads mapped to one barcode, while 56.7% of cells containing reads mapped to multiple barcodes, reflecting a high doublet rate (or RNA contamination?) compared to traditional hashing-based methods. Is such a high doublet rate expected? Any possible explanations?

2. In addition, to prove the usefulness and reliability of their newly developed method, the authors need to demonstrate the technique details, including how many UMIs (mean/median/distribution) from transcribed barcodes are detected per cell, what's the proportion of the top-expressed barcode per cell, what cutoffs are used to decide if a cell can be convincingly assigned to a barcode and what's the percentage of such cells from all sequenced cells.

3. About Figure 1b., I would suggest removing the plasmid graph as the main figure, especially since the texts are too small to see and most of the plasmid components are irrelevant. Instead, the author should submit the plasmid sequence as a supplementary file and draw a schematic of how the sequencing library of the transcribed barcode is constructed.

4. About the annotations of in vitro-derived cell types. The authors applied label transfer using published single-cell RNA-seq data to guide their annotations. However, it's noted that a relatively large fraction of cells seems to have remained "unannotated" (Figure 3c). This is possibly due to an incomplete coverage of cell types in the reference dataset. The authors could try integrating with other publicly available large datasets such as from <https://www.nature.com/articles/s41586-024-07069-w> and examine if the situation can be improved. Moreover, it will be interesting to check whether similar cells following the big broad category can be further broken down into finer cellular states. For example, the authors annotated four forelimb mesoderms clusters (41,42,44,47; Figure 3H), are they corresponding to biologically meaningful substates?

5. Based on Figures 4b and 4c, it's interesting to see that inhibition and activation of BMP pathways give rise to relatively similar results compared to distinct differences between inhibition and activation of Wnt signaling. It would be great for the authors to include more discussions/explanations about this result.

Reviewer #3 (Remarks to the Author):

The manuscript by Shen et al. aims to generate an atlas of multilineage differentiation from pluripotency and leverage the atlas to gain insights into complex human traits. By engineering barcoded iPSCs and using them in a differentiation protocol modulating WNT signaling, the authors seek to showcase the utility of their tool and further test the role of TMEM88 on multilineage differentiation and physiological traits. The authors generated an in vitro atlas that is similar to others in the field, including from the same group (PMID: 30290179), but failed to compare to other in vitro atlases. The authors further discover a regulator of cardiovascular differentiation previously characterized by some of the same authors (TMEM88, PMID: 23924634). The authors then hypothesize potential physiological responses to TMEM88 deletion, and test it using mouse models, but fail to elicit a strong phenotype. Overall, the study is unfocused, lacks scientific rigor and would require major significant revisions to better understand whether the engineered barcoded iPSCs and dataset are valuable resources and how significant the scientific contributions are.

Major

- The authors used cell hashing for their time course experiment, but cell engineered barcodes for perturbing experiment, making the results hard to interpret between the two experiments. Since this study is trying to establish their technology, I would like to see the perturbing experiment repeated with the cell engineered barcodes.
- The rationale to use TRIAGE clustering is not explained well. Please elaborate more on why it was used over traditional cluster annotation. The authors should do a comparison of TRIAGE versus traditional cluster annotation used by others in the field.
- When the “cell peaks” are annotated by TRIAGE, what are the other “non-peak” cells that are not identified by the pipeline? It looks as though most cells are not identified. Are those cells not used in downstream analyses?
- To partially validate the TRIAGE-Cluster peaks, the authors perform pseudotime inference and claim that accuracy is improved when using TRIAGE-Cluster peaks vs all cells. However, the logic behind the pseudotime accuracy metric is flawed. The premise of pseudotime is that there are a range of developmental cell states within each day of differentiation, especially for highly dynamic processes such as early differentiation. Therefore, correlating pseudotime to day of differentiation does not make sense as a pseudotime metric. One could make an argument that since the TRIAGE-Cluster pseudotime analysis is so correlated with day, it is not capturing the heterogeneous cell states known to be present at each day, meaning the TRIAGE-Cluster annotations are less accurate. The authors should perform more thorough comparisons of pseudotime inference between TRIAGE-Cluster peaks vs all cells.

- The authors perform a cell label transfer with several in vivo atlases to help identify and annotate the “cell peaks”, but many of the annotations are different between the various in vivo datasets. What are the final cell peak annotations based on their analyses and can the authors provide gene expression evidence to support the annotations?
- The authors missed a valuable opportunity to utilize their cell engineered barcodes to build a developmental lineage tree by inferring the progenitors of each lineage based on the barcode information. This can then be compared with different developmental trajectory algorithms to benchmark the technology.
- Since the authors are trying to establish this as an in vitro multilineage atlas, more rigorous analysis should be done to extract insights from the atlas, including but not limited to cell-cell interaction and gene regulatory network analyses at each of the different days of differentiation.
- It is unclear how the authors use Summary data-based Mendelian Randomisation (SMR) analysis to narrow down to only TMEM88. FGF10 is another XAV DEG and exhibits higher significance than TMEM88 in Figure 4d, why was that not chosen to test?
- For the SMR analysis, can it be performed in reverse? For example, for the systolic and diastolic blood pressure traits, what are the top enriched genes for each of the tissues profiled? Is TMEM88 one of the top genes?
- The phenotype for the TMEM88 knockout mouse is mild since the systolic and diastolic blood pressure is not significantly different between conditions. Did the authors test other traits on Figure 4f that are associated with TMEM88? Standing height appears to be the most significantly associated based on their analysis.
- Between line 91-102, they mentioned that about 42.0% cells mapped to one, and 56.7% to multiple barcodes. It is clear if only 56.7% cells are unique labeled and can be used for the following analysis. If so, the false negative rate is higher than hashing antibody labeling. The author should show how many barcodes of each original 18 cell lines has, as well as if there is some overlap among these barcodes.

Even though barcode combinations can make each cell with unique labeling, it is not mean one barcode can distinguish them if these cell line share some barcodes.

- Virus infection sometime can cause the iPSC cells differentiation ability changes, AAV labeled iPSCs cell line should be tested in differentiation system to see their differentiation ability difference.
- The authors should describe how they found Wnt related genes. Are they only comparing wnt signal genes between XAV and CHIR group? Or find all differentially expressed genes between XAV and CHIR group? Which stage data they compared? It seems they used day 9 XAV vs CHIR data. As XAV and CHIR were used between Day 3-5, and the cell types can be distinguished already at day 9, the DE genes between XAV and CHIR group at day 9 may not directly related with wnt signal. Day 3-5 is the best time points to find the direct wnt signal related genes that regulate cell differentiation.
- XAV is the downstream inhibitor of wnt signal, it is unclear how XAV can cause other upstream wnt signal genes expression change. The author found Tmem88 through comparing Wnt signal

genes between XAV and CHIR group and claim Tmem 88 is mediators of WNT-requisite cell fate specification. XAV treatment on Tmem 88 KD/KO cells is required for this claim.

- Jitter plot cannot show well the connection between the different cell types and their progenitors, for example, which mesoderm or endoderm population the cardiac or pancreas come from. URD trajectory may be helpful for showing cell fate change gradually with different treatment conditions, and helpful for finding the key genes underlying each treatment condition.

Minor

- Figure 1 does not explain the technology of the cell engineered barcodes well. I would advise drawing a schematic/workflow instead of just showing a plasmid map and expecting the reader to understand how it works. The workflow in Figure 2a is a good example of how to better illustrate the tool.
- The authors often refer to the cell peaks as just “peaks” which is rather confusing since that term is mainly utilized in the epigenomic context. I would advise either stricter use of the term “cell peak” or use a different label.
- The labeling of the genes in the heatmap of Figure 3g are too small.
- Gene expression analysis should be shown for the cell annotations shown in Figure 3h to validate the annotations. Without a heatmap of marker genes or something similar, it is hard to evaluate the annotations.
- The expression of TMEM88 should be shown over the time course of the TMEM88 KD experiment to validate the efficiency and effectiveness of CRISPRi on TMEM88.

Reviewer #4 (Remarks to the Author):

Reviewer #5 (Remarks to the Author):

REVIEWER COMMENTS

Reviewer #1 (Remarks to the Author):

The authors have generated barcoded iPSC lines and performed a WNT-induced differentiation experiment and collected scRNAseq data at 8 time points. They have additionally perturbed WNT, BMP, VEGF signaling along the differentiation trajectory and obtained scRNAseq data at 3 time points. Utilizing their in-house TRIAGE tools, the authors have then annotated the cell types in their atlas and determined the gene programs underlying the annotated cell types. Next, to determine the association between genes and physiological traits, the authors have used Quantitative Trait Loci (QTL) to link their signaling perturbation analyses to physiology. From these analyses, they have identified and further studied the role of *TMEM88* in cardiac physiology and blood pressure, showing that *TMEM88* KO mice have increased variability in systolic and diastolic blood pressure.

Overall, this study generated a lot of high-quality single cell data and analysis, and this differentiation atlas will be a valuable addition to the field. The analyses and techniques used in this study (Millar Pressure-Volume catheter for instance) are reliable methods in the field. Most importantly, given that blood pressure variability is associated with leading causes of death like strokes and myocardial infarctions, the involvement of *TMEM88* in regulation of blood pressure is a fascinating finding for both the advancement of knowledge in the field and in the clinical setting, and merits publication. However, a few concerns would need to be addressed prior to that.

Concerns and suggestions:

1. Figure 4d -- *FZD5* and *TCF7L2* seem similarly important as *TMEM88*. Could the authors elaborate why *TMEM88* was selected? A clearer metric to determine *TMEM88* as a top factor, or testing a few more of the top candidates using the SMR analysis and expression in the atlas will be helpful here.

For selecting the gene to follow up, our aim was to split the difference between highly significant differential expression and significant association with a potentially promising phenotype from SMR analysis, so as to choose a gene with some potential novelty. In comparison to *TMEM88*, *FZD5* and *TCF7L2* are quite well characterised with regards to their role in the Wnt signalling pathway as well as during development¹⁻⁷, leading us to favour investigation into *TMEM88*.

The reviewer's point, however, is well taken. We have now provided the SMR analysis and gene expression UMAPs of three other high-scoring WNT-related genes from **Figure 4d** (*FZD5*, *TCF7L2*, and *NDRG2*) (**Figure 4e-f**, **Figure S4c**) and amended the text in the results (**page 9**) to evaluate all four candidate genes and clarify the intent behind selecting a gene for further study. Indeed, this expanded SMR analysis shows that *TCF7L2* expression in the aorta has many highly significant associations with primarily anthropometric traits such as hip circumference, as well as the same blood pressure traits associated with *TMEM88* expression. While these associations with *TCF7L2* expression in the aorta specifically is surprising, the enrichment of these anthropometric traits seems less so, given its well-studied role in type 2 diabetes⁸.

2. Could the authors provide a graph of the number of cells for each of the treatment conditions at the 3 time points (Figure S1d) (Similar to the graph in S1C top left). This will be essential in evaluating the data presented in Figure 4a where %contribution per condition is used as a metric. It may be that considerably fewer cells are captured in some of the treatment conditions (especially at d9), which would bias the inferences.

We have added this graph to **Figure S1g**, however the percent contributions displayed in **Figure 4a** are scaled to each condition, so the size of each point indicates percentage of total cells from each treatment at day 9 that contribute to each cell type. To clarify this point, we have also amended the corresponding **Figure 4a** figure legend.

3. Figure 4a – Could the authors allude to the reasons why low levels of Wnt inhibition seems to favor the pancreas, liver, tracheal progenitor lineages?

Since these lineages also seem to be favoured in the control groups (**Figure 4a**), we believe this to be aligned with the default outcome of the mesendoderm differentiation protocol. This suggests that the lower concentration of XAV was perhaps too low to induce an effect on differentiation (compared to the high XAV group). For the VEGF, the same could be the case, though studies show that VEGF signalling has a more prominent role at later points in mesendoderm-derivative differentiation for vascular specification⁹, but less so immediately following gastrulation as tested in this study. We have now included discussion to this effect in the text on **page 8**.

4. Also, in Figure 4a, both the inhibition and promotion of BMP signaling seems to contribute to the same cell types. Could the authors explain why this could be the case? Additionally, I am curious if this could be because using a Wnt-activation based differentiation protocol would overshadow the perturbations from other signaling pathways?

We agree that this is somewhat unexpected. Our hypothesis is that a BMP inhibitory mechanism is being enacted in the cells following the treatment between days 3 and 5 of differentiation so that by day 9, the BMP4-treated cells end up resembling the groups treated with the BMP inhibitors. This seems possible considering the transient formation of more posterior foregut cell types in the BMP4 cells at day 5, which is more in line with the known posteriorising effect of BMP4 in embryogenesis, an effect that does not persist until day 9 of this differentiation (**Figure 4a**). We have updated the text on **page 8** to suggest this, but more substantial follow-up work would be required to substantiate this claim, which we see to be outside the scope of the study.

Whether this is due to the WNT activation-based protocol is an astute question. Some studies¹⁰ show the CHIR-induced WNT activation alone tends to favour definitive endoderm and its foregut derivatives as the “default” differentiation outcome, from which these BMP treatments appear to deviate. This could indicate that the BMP modulation is not completely overshadowed by the CHIR activation, though future studies would also be required to compare their effects with different mesendoderm induction protocols.

5. Figure 3h works great to showcase the data, and it’s remarkable to see the lineage specification in the cells captured as the differentiation timepoints pass. However, page 7, line 161-175 could use more details and elaboration before an integrated view can be presented (line 77). Especially for readers unfamiliar with TRIAGE tools, some more details or stepwise logic of Figure 3e-g will be helpful.

We thank the reviewers for their feedback. Based on comments from all reviewers regarding our analysis methods, we have made changes to **Figures 2-3** to better justify the data interpretation strategy. We have amended the text on **pages 6-7** to better convey this point, as well as to clarify the differences between the three *TRIAGE* tools, their individual purposes, and how they contribute to our final annotations.

6. If *TMEM88* KD reduced differentiation of cardiac cells in vitro, it seems surprising that the *TMEM88* KO mice (especially homozygous) show no impact on cardiac structure and function (line 238). Could the authors elaborate on possible reasons for this?

The lack of a distinct cardiac defect in the *Tmem88* KO mice could be due to a range of reasons, but a likely possibility is a compensatory mechanism. This would be in line with a proposition that *Tmem88* functions as a fine-tuning regulator¹¹, rather than a central driver for specification. Supporting this hypothesis, a recent study¹² showed that *Tmem88a* (orthologous to the human *TMEM88*) only showed sustained developmental defects when knocked out in conjunction with paralogue *Tmem88b*. This could suggest a possible compensatory mechanism for *Tmem88* LOF in our mouse model. We have adjusted the text on **page 11** to discuss this in the manuscript.

Minor:

1. Table S1 seems crucial to understand the barcoding strategy and experimental outline of the perturbation. My recommendation would be to include it as a panel in Figure 2, or as a main table. This would also avoid the confusion whether each cell line has 1 barcode or 1 full barcode library (containing 10k barcodes) (line 95). Another option could be to move the panel 2b to supplementary or combine it with the Table S1.

We appreciate this feedback and have now included **Table S1** as main **Table 1**.

2. Figure 1a – The barcoding vs cell vs cost diagram is a bit confusing. Barcoding at single cell level can also include population level barcodes in the cassette, and could also be done at the start of the experiment. Could the authors elaborate what the intended message was in this figure 1a panel?

This panel was intended to compare some aspects of not using sample multiplexing (right) and current post-experimental sample multiplexing strategies (left). The former has one barcode per cell and only one biological sample per sequencing reaction, and the latter incorporates sample barcodes where there are more cells sharing the same barcode, there are fewer (sample) barcodes overall, but associated reagent costs per sample are higher. We were using this comparison to highlight benefits of genomic barcoding strategies that allow for more sample barcodes with lower costs per sample. That being said, we agree that this schematic is not the most effective at conveying these points and since the specific benefits of our genomic barcoding approach are already better discussed in the text, we have chosen to remove this panel altogether. A new **Figure 1a** schematic now focuses entirely on the barcoding method which is the most important aspect for setting up the experimental work.

3. Figure 2a – “Barcode cell editing by CRISPR/Cas9” – should this more aptly say: “Barcode insertion into the cell lines by CRISPR/Cas9” to avoid confusion since the barcodes aren’t being edited?

We have changed the text to say “Barcode insertion by CRISPR/Cas9” in **Figure 2a**.

4. Figure 4b-c legend: Should be (b) and (c) instead of the (c) and (d)

We have fixed the **Figure 4** legend accordingly.

Reviewer #2 (Remarks to the Author):

In this study, Shen et. al constructed genetically barcoded iPSCs for sample multiplexing in single-cell RNA-seq experiments and used them to study the differentiation of iPSCs upon standard and multiple drug treatment conditions. Extensive computational analyses were applied to map the identity of differentiated cells, for characterizing the genetic programs driving different lineages and the outcome of each perturbation. With the platform, the authors identified TMEM88 as a putative regulator affecting cardiovascular cell differentiation and related that to cardiovascular and anthropometric traits using gene-traits association analysis, and explored the physiological role of TMEM88 in regulating blood pressure in vivo using KO mice. Taken together, the study represents a good case study of how one can couple iPSC differentiation models, single-cell RNA-seq, and computation tools to pinpoint novel gene targets of human diseases and cell types affected, with the potential to link genes, cell types, and human diseases in a broad context. The following are several comments that should be fixed before the publication of the manuscript:

1. More technical justifications are required for evaluating the performance of single-cell capture of transcribed barcodes. For example, in line 100, the author mentioned that only 42.0% of cells containing reads mapped to one barcode, while 56.7% of cells containing reads mapped to multiple barcodes, reflecting a high doublet rate (or RNA contamination?) compared to traditional hashing-based methods. Is such a high doublet rate expected? Any possible \?

We thank the reviewers for this feedback. We have now included a more thorough breakdown of the barcoding metrics used to derive these percentages (**Table S1, Figure 1h-i, Figure S1a**). The example mentioned is related to the pilot scRNA-seq experiment we conducted as a proof-of-concept for the barcoded cell lines. This dataset had a very low sequencing depth for the genomic barcode library, so we opted to use the barcode with the highest number of reads to assign each cell's sample of origin and "doublets" were cells with exactly the same number of reads for two barcodes. As such, the 56.7% of cells with reads mapping to multiple barcodes includes cells with even one read mapping to a second barcode and indeed in **Table S1 and Figure 1h** we see that on average cells only have reads mapping to two barcodes, and most of the time the second most highly "expressed" barcode has only one mapped barcode. **Table S1** also shows that while the hashed cells don't typically have reads mapped to a second barcode, this is likely due to the even lower sequencing depth in the hashing library.

As the low sequencing depth in the barcoding and hashing libraries from the pilot study render these metrics difficult to assess, we have additionally provided a similar set of metrics for the larger atlas dataset (**Table S2**). We compare the hashed time course reference dataset to the three genomically barcoded libraries from the signalling dataset (**pages 5-6, Table S2, Figure S1b**) and also specifically highlight the samples that overlap between the hashed and barcoded libraries (control cells from days 2, 5, and 9) in **Table S2**. With the increased sequencing depth in these libraries, we observe that both our method and cell hashing similarly have stray reads mapped to almost all the other possible barcodes (**Figure S1b**), but while the genomic barcodes have a lower percentage of reads assigned to the most highly expressed barcode in each cell (53.2% in genomic barcodes versus 87.5% in cell hashing), the doublet rate is actually comparable between the methods (**Figure S1b, Table S2**).

2. In addition, to prove the usefulness and reliability of their newly developed method, the authors need to demonstrate the technique details, including how many UMIs (mean/median/distribution) from transcribed barcodes are detected per cell, what's the proportion of the top-expressed barcode per cell, what cutoffs are used to decide if a cell can be convincingly assigned to a barcode and what's the percentage of such cells from all sequenced cells.

We have added two supplementary tables to provide these additional metrics for the pilot barcoding study (**Table S1**) and the atlas dataset (**Table S2**), as well as visualisations of select metrics in **Figure 1h-i** and **Figure S1a-b**. As mentioned in our response to the previous point, no cut-offs were employed in the pilot barcoding dataset as it was primarily designed to determine whether these internally transcribed barcodes could be captured in scRNA-seq. For both the hashed and genomically barcoded libraries in the atlas dataset, we used the *HTODemux* function provided as part of the Seurat package with the default 0.99 quantile cutoff (described in the **Methods**) to distinguish the level of reads for each barcode in each cell from the background

ambient noise. Percentages of cells assigned as “Negative”, “Doublet” and “Singlet” based on *HTODemux* are also provided in **Table S2**.

3. About Figure 1b., I would suggest removing the plasmid graph as the main figure, especially since the texts are too small to see and most of the plasmid components are irrelevant. Instead, the author should submit the plasmid sequence as a supplementary file and draw a schematic of how the sequencing library of the transcribed barcode is constructed.

We thank the reviewers for the suggestion. We have now provided the annotated plasmid sequence in **Supplementary File 1** and replaced the previous **Figure 1a-b** with a schematic as recommended (**Figure 1a**).

4. About the annotations of in vitro-derived cell types. The authors applied label transfer using published single-cell RNA-seq data to guide their annotations. However, it's noted that a relatively large fraction of cells seems to have remained “unannotated” (Figure 3c). This is possibly due to an incomplete coverage of cell types in the reference dataset. The authors could try integrating with other publicly available large datasets such as from <https://www.nature.com/articles/s41586-024-07069-w> and examine if the situation can be improved. Moreover, it will be interesting to check whether similar cells following the big broad category can be further broken down into finer cellular states. For example, the authors annotated four forelimb mesoderms clusters (41,42,44,47; Figure 3H), are they corresponding to biologically meaningful substates?

Considering the reviewers points in this comment, we have carefully re-evaluated and made minor adjustments to cell type annotations in **Figure 3**. We agree with the reviewer that the “unannotated” cells remaining after the three rounds of label transfer could be due to incomplete coverage of cell types in the reference datasets, or even cell types that aren't represented *in vivo* at all. It is for this reason that we performed label transfer on multiple in vivo data sets (**Figure 3a-d**) such that multiple reference data sets are used to inform annotations thereby strengthening justification for cell type interpretation.

Heeding the reviewers' advice, we have also now repeated the label transfer analysis using the suggested reference point from Qiu *et al.*, (**Figure S2d**), however this unfortunately resulted in markedly low mapping confidence scores and mapped the cells primarily as either “Neural Crest” or “Gut” cells. It is possible that this is due to the mismatch in timing between the query and reference datasets, with the epiblast and early mesendodermal cell types in our dataset likely to arise before the earliest E8 timepoint in the Qiu *et al.*, dataset.

With regards to the groups of similar cell type cluster annotated with shared annotations (such as the forelimb mesoderm clusters), this is a very common issue in single cell analysis where the same cell types are annotated across the sub-clusters¹³. While it is possible that these clusters represent different cell subtypes, due to their highly similar identity-defining genes and enriched gene ontologies (**Figure 4e** (dendrogram) & **Tables S6-8**) it seems more likely that these cell clusters arise only from the way the cells distributed in 2D space from the dimensionality reduction. Indeed, as discussed previously¹⁴, single cell clustering methods are susceptible to artifacts produced by the dimensionality reduction method that make it challenging to distinguish cell subtypes depending on the resolution of clustering chosen.

5. Based on Figures 4b and 4c, it's interesting to see that inhibition and activation of BMP pathways give rise to relatively similar results compared to distinct differences between inhibition and activation of Wnt signaling. It would be great for the authors to include more discussions/explanations about this result.

We agree that this is somewhat unexpected and this point was also raised by another reviewer, wondering whether this is due to the WNT activation-based differentiation protocol. Some studies¹⁰ show the CHIR-induced WNT activation alone tends to favour definitive endoderm and its foregut derivatives as the “default” differentiation outcome, from which these BMP4 treatments appear to deviate. This could indicate that the BMP4 modulation is not completely overshadowed by the CHIR activation, though future studies would also be required to compare their effects with different mesendoderm induction protocols.

Our hypothesis is that a BMP inhibitory mechanism is being enacted in the cells following the treatment between days 3 and 5 of differentiation so that by day 9, the BMP4-treated cells end up resembling the groups treated with the BMP inhibitors. This seems possible considering the transient formation of more posterior

foregut cell types in the BMP4 cells at day 5, which is more in line with the known posteriorising effect of BMP4 in embryogenesis, an effect that does not persist until day 9 of this differentiation (**Figure 4a**). We have updated the text on **page 8** to suggest this, but more substantial follow-up work would be required to substantiate this claim, which we see to be outside the scope of the study.

Reviewer #3 (Remarks to the Author):

The manuscript by Shen et al. aims to generate an atlas of multilineage differentiation from pluripotency and leverage the atlas to gain insights into complex human traits. By engineering barcoded iPSCs and using them in a differentiation protocol modulating WNT signaling, the authors seek to showcase the utility of their tool and further test the role of TMEM88 on multilineage differentiation and physiological traits. The authors generated an in vitro atlas that is similar to others in the field, including from the same group (PMID: 30290179), but failed to compare to other in vitro atlases. The authors further discover a regulator of cardiovascular differentiation previously characterized by some of the same authors (TMEM88, PMID: 23924634). The authors then hypothesize potential physiological responses to TMEM88 deletion, and test it using mouse models, but fail to elicit a strong phenotype. Overall, the study is unfocused, lacks scientific rigor and would require major significant revisions to better understand whether the engineered barcoded iPSCs and dataset are valuable resources and how significant the scientific contributions are.

Major

- The authors used cell hashing for their time course experiment, but cell engineered barcodes for perturbing experiment, making the results hard to interpret between the two experiments. Since this study is trying to establish their technology, I would like to see the perturbing experiment repeated with the cell engineered barcodes.

We recognise that using different multiplexing methods for the two datasets can complicate interpretation of barcoding efficacy between the two experiments. We have however anticipated this issue and designed the experiments to have overlapping samples. Barcodes BC07, BC08, BC12, and BC13 also underwent cell hashing in the pilot study (**Figure 1f-j**), and in the atlas dataset, the control/untreated cells at days 2, 5, and 9 of differentiation are represented in both the cell hashing and engineered barcoding datasets (**Figure 2a**). See the new **Table S1** and **Table S2** where we directly compare a range of barcoding metrics between the two methods and for these overlapping samples specifically. In particular, we show that the resulting transcriptomic quality and negative/doublet rate is similar between barcoded and hashed lines.

- The rationale to use TRIAGE clustering is not explained well. Please elaborate more on why it was used over traditional cluster annotation. The authors should do a comparison of TRIAGE versus traditional cluster annotation used by others in the field.

We thank the reviewers for this feedback. We have now included the results and analysis of whole-dataset clustering using Seurat in **Figure 2e**, **Figure S2a** & **Figure 3a** to provide a more traditional approach to cell type identification and annotation of the atlas dataset, strengthening it as a resource for the field. We have also now amended the text on **pages 6-7** to provide a clearer rationale for using *TRIAGE-Cluster* as a method to supplement whole-dataset clustering to provide a unique additional perspective into cell subtype identification, as well as highlight a comparison of the information captured by these two clustering methods provided in **Figure S3b**.

- When the “cell peaks” are annotated by TRIAGE, what are the other “non-peak” cells that are not identified by the pipeline? It looks as though most cells are not identified. Are those cells not used in downstream analyses?

The reviewer has correctly interpreted these methods. In our previous study developing *TRIAGE-Cluster*¹⁴, we demonstrate that compared to benchmark clustering methods such as *Seurat*, this method more effectively identifies cell populations in the data set that are biologically distinct (have a reduced pairwise Spearman correlation of their pseudo-bulk gene expression profiles across all cell types at equivalent clustering resolutions). While many cells are excluded, the enriched cell types provide an effective approach to interpret cell type diversity. To help address the reviewers concern and provide readers with data relevant to this study, we have provided the same analysis to compare cell type diversity using *Seurat* vs *TRIAGE-Cluster* in **Figure S3a-b**. The data show that whether using all genes or only highly variable genes, the use of *TRIAGE* methods for data interpretation results in a significant reduction in gene expression correlation between the clusters in the data set. In addition to providing these data, we acknowledge the challenge of interpreting results of the

study for readers not familiar with *TRIAGE-Cluster*¹⁴, and have therefore added new text throughout the results to provide clearer justification and explanation of the methods in the context of interpreting the results.

- To partially validate the *TRIAGE-Cluster* peaks, the authors perform pseudotime inference and claim that accuracy is improved when using *TRIAGE-Cluster* peaks vs all cells. However, the logic behind the pseudotime accuracy metric is flawed. The premise of pseudotime is that there are a range of developmental cell states within each day of differentiation, especially for highly dynamic processes such as early differentiation. Therefore, correlating pseudotime to day of differentiation does not make sense as a pseudotime metric. One could make an argument that since the *TRIAGE-Cluster* pseudotime analysis is so correlated with day, it is not capturing the heterogeneous cell states known to be present at each day, meaning the *TRIAGE-Cluster* annotations are less accurate. The authors should perform more thorough comparisons of pseudotime inference between *TRIAGE-Cluster* peaks vs all cells.

The reviewer has made an excellent point about pseudotime being suboptimal for evaluating the *TRIAGE-Cluster* cell types. Based on this comment and this reviewer's additional recommendations (below) to expand the scope of data analysis to better justify the impact of the data resource, we decided to remove all trajectory analysis from the study. We also de-emphasise *TRIAGE-Cluster* outputs in favour of traditional whole-dataset clustering for highlighting the most distinct cell types (**Figure S3b**) for simplified characterisation of cell type heterogeneity in single-cell data. Collectively, the recommendations by this reviewer have significantly improved the study by strengthening the cohesiveness of the narrative from a cell and biological perspective rather than convoluting the results by justifying analysis methods that have already been the focus of previous publications.

- The authors perform a cell label transfer with several in vivo atlases to help identify and annotate the “cell peaks”, but many of the annotations are different between the various in vivo datasets. What are the final cell peak annotations based on their analyses and can the authors provide gene expression evidence to support the annotations?

The final cell peak annotations based on the collective outputs of multiple analyses including the label transfer from 4 different in vivo reference data sets (**Figure 3a-d**), gene modules describing each cell type peak (*TRIAGE-Parser* analysis), and marker genes. We have now also provided a bubble plot of select marker genes for each of the cell type peaks selected from the most differentially expressed genes as well as the top 100 identity-defining genes for each cell type based on the *TRIAGE* analysis (**Figure S3e**). These top 100 genes were the input for *TRIAGE-Parser* analysis, and we have now also provided the full list of genes for each peak in each **Table S6** to support the final annotations. Based on these new data, we have thoroughly re-evaluated the cell type annotations and made minor adjustments to ensure they are the most biologically justified interpretation (see **Figure 3h** jitter plot).

- The authors missed a valuable opportunity to utilize their cell engineered barcodes to build a developmental lineage tree by inferring the progenitors of each lineage based on the barcode information. This can then be compared with different developmental trajectory algorithms to benchmark the technology.

The reviewer makes a good recommendation based on many other barcoding methods that have been used for *in vitro* lineage tracing^{15,16}. Unfortunately, the design of our barcoding strategy does not enable this analysis. We use a single unchanging barcode in each of the cell lines, unlike the mutable or combinatorial barcodes used in some high-throughput lineage tracing methods¹⁷. In our barcoding method, each barcoded sample is derived from a cell population at one time point of differentiation, therefore there is no capacity to determine lineage relationships over time. These barcoded lines are only suited for sample multiplexing, or co-culture and transplantation experiments. Noting the importance of the reviewer's comment which will likely be shared by readers familiar with diverse barcoding methods, we have provided a more explicit description of the strengths and limitations of our barcoding method, see manuscript **page 11**.

- Since the authors are trying to establish this as an in vitro multilineage atlas, more rigorous analysis should be done to extract insights from the atlas, including but not limited to cell-cell interaction and gene regulatory network analyses at each of the different days of differentiation.

We thank the reviewer for the suggestions. We have now provided a significant expansion of analysis outputs for the cell types identified from traditional, whole-dataset clustering (**Figure 2d**) to provide more characterisation for users of the atlas (**Figure S2**). This includes evaluation of differentially expressed marker genes for each cell type (**Figure S2a**), enrichment of GO terms and KEGG pathways (**Figure S2b-c**), label transfer analysis against mouse and human *in vivo* reference data (**Figure S2d**), cell-cell interaction prediction analysis compared between treatment groups over time (**Figure S2e**, **Table S3** and **Supplementary File 2**), enrichment of gene regulon activity in each cell type (**Figure S2f** and **Table S4**), and URD differentiation trajectory reconstruction to assess relationships between clusters (**Figure S2g**), as suggested by the reviewer in a comment below.

Overall, the DE genes, GO/KEGG enrichment, label transfer, and gene regulon activity enrichment provide further support for the cell type annotation and align with each other well (**Figure S2a-d, f**). The most significant predicted cell-cell interactions were those related to structural features and extracellular matrix, such as collagen and laminin interactions. Focusing on WNT and BMP signalling on day 5, we highlight signalling relationships between cell types that are shared and distinct between treatment groups (**Figure S2e**). The trajectory analysis also serves to support some of the cell annotations, successfully separating the broad endoderm and mesoderm lineages into different branches aligning with our expectations, with the exception of the axial mesoderm, which is included in the same branch as the anterior foregut endoderm (**Figure S2g**).

As these characterisation analyses have been added in the middle of the greater study for which we have substantial downstream analysis and data, we felt that incorporation of detailed evaluation of the results from each of these analyses would be detrimental to the overall flow of the narrative and impact on word count limits of the study. For this reason, we have incorporated relevant findings throughout the manuscript in selected contexts and placed all results in the supplemental content for interested readers.

- It is unclear how the authors use Summary data-based Mendelian Randomisation (SMR) analysis to narrow down to only *TMEM88*. *FGF10* is another XAV DEG and exhibits higher significance than *TMEM88* in **Figure 4d**, why was that not chosen to test?

For selecting the gene to follow up, our aim was to split the difference between highly significant differential expression and significant association with a potentially promising phenotype from SMR analysis, so as to choose a gene with some potential novelty. As the SMR was conducted after the differential expression analysis, and not given much weight for selection of a candidate, *FGF10* was excluded on the basis of its lower differential expression significance and well-studied relationship to the WNT pathway during lineage specification¹⁸⁻²¹. Furthermore, in comparison to *TMEM88*, *FZD5* and *TCF7L2* are quite well characterised with regards to their role in the Wnt signalling pathway as well as during development¹⁻⁷, leading us to favour investigation into *TMEM88*.

The reviewer's point, however, is well taken. We have now provided the SMR analysis and gene expression UMAPs of three other high-scoring WNT-related genes from **Figure 4d** (*FZD5*, *TCF7L2*, and *NDRG2*) (**Figure 4e-f**, **Figure S4c**) and amended the text in the results (**page 9**) to evaluate all four candidate genes and clarify the intent behind selecting a gene for further study. Indeed, this expanded SMR analysis shows that *TCF7L2* expression in the aorta has many highly significant associations with primarily anthropometric traits such as hip circumference, as well as the same blood pressure traits associated with *TMEM88* expression. While these associations with *TCF7L2* expression in the aorta specifically is surprising, the enrichment of these anthropometric traits seems less so, given its well-studied role in type 2 diabetes⁸.

- For the SMR analysis, can it be performed in reverse? For example, for the systolic and diastolic blood pressure traits, what are the top enriched genes for each of the tissues profiled? Is *TMEM88* one of the top genes?

SMR analysis can be performed for all genes in the genome with eQTLs. We have now performed this and provided the top 100 enriched genes for systolic and diastolic blood pressure, as well as specifically pulling out associations of *TMEM88* and the other differentially expressed WNT-related genes from **Figure 4d** and included these results in **Table S8**. While *TMEM88*'s association does not rank highly in the context of all genes with associations to blood pressure, most WNT-related genes do not appear to rank so highly (**Table**

S9). *TMEM88* however is fairly distinct in its increased significance and larger negative effect on both diastolic and systolic blood pressure in the two cardiac tissues.

- The phenotype for the *TMEM88* knockout mouse is mild since the systolic and diastolic blood pressure is not significantly different between conditions. Did the authors test other traits on Figure 4f that are associated with *TMEM88*? Standing height appears to be the most significantly associated based on their analysis.

Since very little is known about *TMEM88* in development, we chose to address this concern by performing a new unsupervised phenotyping of the animals using micro-computed tomography (micro-CT) to assess global organ morphometry in embryos. This provides an unsupervised means to evaluate the associations related to diverse anthropometric traits from the SMR analysis (**Figure S5h**). Of note, both the left and right pelvic girdles showed significantly increased volume ratios in the mutant animals, aligning with the SMR results indicating a negative association between *TMEM88* and hip circumference. For the standing height phenotype, the micro-CT scans indicate increased size of the lumbar, thoracic, and cervical vertebrae in the mutant embryos (**Figure S5h**), which aligns with the association of *TMEM88* expression in the atrial appendage. Together with vertebral size not being a direct measure of standing height, these factors highlight the importance of *in vivo* validation of associations identified from statistical genetics analyses. Collectively, our novel *Tmem88* KO mouse coupled with these results provide opportunities for further study into the mechanisms of *TMEM88* in diverse organ systems and physiological contexts.

- Between line 91-102, they mentioned that about 42.0% cells mapped to one, and 56.7% to multiple barcodes. It is clear if only 56.7% cells are unique labeled and can be used for the following analysis. If so, the false negative rate is higher than hashing antibody labeling. The author should show how many barcodes of each original 18 cell lines has, as well as if there is some overlap among these barcodes. Even though barcode combinations can make each cell with unique labeling, it is not mean one barcode can distinguish them if these cell line share some barcodes.

The nature of this method is that a single 15bp genomic barcode is incorporated into each of the 18 cell lines so that there are 18 barcoded lines with a different barcode each. As such, our method is directly analogous to cell hashing, where the transcripts from the genomic barcode (like the cell hashing oligos) bind to the 10x cellular barcodes and can be treated as an expressed “gene” in each cell. In this way, we would expect each cell to have reads mapping primarily to a single barcode, which is used to identify its sample of origin. We have amended the text on **page 5** and updated **Figure 1a** (as per the reviewer’s suggestion below) to better explain the barcode construction, sequencing, and mapping strategy of our method.

For the metrics on line 91-102, by referencing the 42% of cells mapping to one barcode, the intention was to describe the percentage of singlet cells based identified by the barcoding libraries. We have however revisited this metric and from looking at the distribution of reads in each cell (**Figure 1h-i**, **Figure S1b**), we show that while 56.7% of cells have reads mapping to more than one barcode (potential multiplets), the majority of cells have a large enough difference in number of reads mapping to the dominant barcode in each cell, such that the remaining reads can be disregarded as noise.

- Virus infection sometime can cause the iPSC cells differentiation ability changes, AAV labeled iPSCs cell line should be tested in differentiation system to see their differentiation ability difference.

We acknowledge that many cell barcoding methods²² use viruses for delivery of the barcodes. Though this is common approach when using barcode libraries, this is not the case for our method. We performed CRISPR gene editing using Cas9 and plasmid transfections to integrate the genomic barcodes into the *AAVS1* safe harbour locus on chromosome 19. Correctly edited cell lines were selected by puromycin. Studies have shown that transgene incorporation into this locus shows no adverse effects on cell phenotype and differentiation capability^{23,24}. The amended text on **page 5** and new **Figure 1a** schematic have been significantly revised to clarify the confusion around this point.

In addition, to ensure the results of the barcoding were robust, each unique signalling treatment-timepoint combination in the atlas dataset was replicated with two different barcoding lines to demonstrate high

transcriptional similarity between independent cell lines (**Figure S1g**). This similarity supports comparable differentiation capabilities across lines for the relevant cell types at matched timepoints and treatment groups.

- The authors should describe how they found Wnt related genes. Are they only comparing wnt signal genes between XAV and CHIR group? Or find all differentially expressed genes between XAV and CHIR group? Which stage data they compared? It seems they used day 9 XAV vs CHIR data. As XAV and CHIR were used between Day 3-5, and the cell types can be distinguished already at day 9, the DE genes between XAV and CHIR group at day 9 may not directly related with wnt signal. Day 3-5 is the best time points to find the direct wnt signal related genes that regulate cell differentiation.

The reviewer makes a reasonable point that day 3-5 time points would be most direct approach to evaluate the acute effects of Wnt perturbation mechanisms. However, our interest in identifying Wnt-associated genes was not restricted to only those responding to the drug perturbations, but more-so to identify Wnt-associated genes involved across all cell lineages and time points resulting from Wnt perturbations. The results identified at day 9 provided the most biologically interesting results, allowing us to functionally link gene expression differences in vitro to physiological traits and phenotypes in vivo. To address this concern, we amended the text on **page 9** to clarify that we performed differential gene expression between the XAV and CHIR groups at day 9 and used gene ontology annotations to highlight genes with WNT-related annotations.

- XAV is the downstream inhibitor of wnt signal, it is unclear how XAV can cause other upstream wnt signal genes expression change. The author found Tmem88 through comparing Wnt signal genes between XAV and CHIR group and claim Tmem 88 is mediators of WNT-requisite cell fate specification. XAV treatment on Tmem 88 KD/KO cells is required for this claim.

The role for *TMEM88* as a Wnt inhibitor is currently its only described mechanism. However, the PDZ binding motif at the C-terminus of *TMEM88* which acts to inhibit Wnt pathway activity may also interact with diverse other proteins or mechanisms not yet described. Thousands of human proteins are predicted to contain PDZ-binding motifs. For example, the ELM (Eukaryotic Linear Motif) resource, which catalogues protein motifs, contains data on a significant number of such motifs in its database. Additionally, studies utilizing the proteomes of model organisms, including humans, estimate that between 1-3% of proteins may contain PDZ-binding motifs. Therefore, the expression response of *TMEM88* associated with Wnt inhibition is likely more nuanced. We also highlight that previous studies^{11,25} have already been performed using in vitro and in vivo analysis showing that Wnt inhibition treatment partially rescues cardiac lineage-associated defects caused by *TMEM88* KD. These studies highlight that *TMEM88* likely has many more roles in governing cell signalling other than its known role mediating Wnt pathway activity. To address this concern, we have added further discussion on **page 11**.

- Jitter plot cannot show well the connection between the different cell types and their progenitors, for example, which mesoderm or endoderm population the cardiac or pancreas come from. URD trajectory may be helpful for showing cell fate change gradually with different treatment conditions, and helpful for finding the key genes underlying each treatment condition.

The dendrogram in **Figure 3h** is designed to show similarities in the regulatory genes underpinning each cell, and we chose to use it in this panel to match the cell type order in **Figure 3e**. The reviewer's suggestion of using URD trajectory analysis is well taken and as such, we have now employed it to assess ancestor-descendant relationships between cell types from the traditional clustering (**Figure S2g**).

Minor

- Figure 1 does not explain the technology of the cell engineered barcodes well. I would advise drawing a schematic/workflow instead of just showing a plasmid map and expecting the reader to understand how it works. The workflow in Figure 2a is a good example of how to better illustrate the tool.

We have now moved the plasmid map to **Supplementary File 1** and replaced **Figure 1a-b** with a workflow of the engineered barcodes in the new **Figure 1a** as advised.

- The authors often refer to the cell peaks as just “peaks” which is rather confusing since that term is mainly utilized in the epigenomic context. I would advise either stricter use of the term “cell peak” or use a different label.

We thank the reviewers for their advice on this point and we have corrected the text so that all references to the cell peaks are described as “cell type peaks” or “cell peaks”.

- The labeling of the genes in the heatmap of Figure 3g are too small.

To address this concern, we have removed the gene labels altogether and instead provided them for each cell type peak with gene cluster annotations in **Table S6**. We have also provided the corresponding GO terms for each cell type peak’s gene clusters in **Table S8**.

- Gene expression analysis should be shown for the cell annotations shown in Figure 3h to validate the annotations. Without a heatmap of marker genes or something similar, it is hard to evaluate the annotations.

We have now added a bubble plot of marker genes for each of the *TRIAGE* cell type peaks to **Figures S3e** to support the selected annotations.

- The expression of *TMEM88* should be shown over the time course of the *TMEM88* KD experiment to validate the efficiency and effectiveness of CRISPRi on *TMEM88*.

We have added data showing the number of cells expressing *TMEM88* over time in each of the three experimental groups in **Figure 5d**.

References

1. Kemp, C.R., Willems, E., Wawrzak, D., Hendrickx, M., Agbor Agbor, T., and Leyns, L. (2007). Expression of Frizzled5, Frizzled7, and Frizzled10 during early mouse development and interactions with canonical Wnt signaling. *Dev Dyn* 236, 2011-2019. 10.1002/dvdy.21198.
2. Poulain, M., and Ober, E.A. (2011). Interplay between Wnt2 and Wnt2bb controls multiple steps of early foregut-derived organ development. *Development* 138, 3557-3568. 10.1242/dev.055921.
3. Yang, A., Chidiac, R., Russo, E., Steenland, H., Pauli, Q., Bonin, R., Blazer, L.L., Adams, J.J., Sidhu, S.S., Goeva, A., et al. (2024). Exploiting spatiotemporal regulation of FZD5 during neural patterning for efficient ventral midbrain specification. *Development* 151. 10.1242/dev.202545.
4. Chodelkova, O., Masek, J., Korinek, V., Kozmik, Z., and Machon, O. (2018). Tcf7L2 is essential for neurogenesis in the developing mouse neocortex. *Neural Development* 13, 8. 10.1186/s13064-018-0107-8.
5. Faro, A., Boj, S.F., Ambrósio, R., van den Broek, O., Korving, J., and Clevers, H. (2009). T-cell factor 4 (tcf7l2) is the main effector of Wnt signaling during zebrafish intestine organogenesis. *Zebrafish* 6, 59-68. 10.1089/zeb.2009.0580.
6. Miao, D., Ren, J., Jia, Y., Jia, Y., Li, Y., Huang, H., and Gao, R. (2024). PAX1 represses canonical Wnt signaling pathway and plays dual roles during endoderm differentiation. *Cell Commun Signal* 22, 242. 10.1186/s12964-024-01629-3.
7. Zhang, X., Chen, Y., Ye, Y., Wang, J., Wang, H., Yuan, G., Lin, Z., Wu, Y., Zhang, Y., and Lin, X. (2017). Wnt signaling promotes hindgut fate commitment through regulating multi-lineage genes during hESC differentiation. *Cell Signal* 29, 12-22. 10.1016/j.cellsig.2016.09.009.
8. del Bosque-Plata, L., Martínez-Martínez, E., Espinoza-Camacho, M.Á., and Gragnoli, C. (2021). The Role of TCF7L2 in Type 2 Diabetes. *Diabetes* 70, 1220-1228. 10.2337/db20-0573.
9. Haigh, J.J. (2008). Role of VEGF in organogenesis. *Organogenesis* 4, 247-256. 10.4161/org.4.4.7415.
10. Hoepfner, J., Kleinsorge, M., Papp, O., Ackermann, M., Alfken, S., Rinas, U., Solodenko, W., Kirschning, A., Sgodda, M., and Cantz, T. (2016). Biphasic modulation of Wnt signaling supports efficient foregut endoderm formation from human pluripotent stem cells. *Cell Biology International* 40, 534-548. <https://doi.org/10.1002/cbin.10590>.
11. Novikov, N., and Evans, T. (2013). Tmem88a mediates GATA-dependent specification of cardiomyocyte progenitors by restricting WNT signaling. *Development* 140, 3787-3798. 10.1242/dev.093567.
12. Zhang, M., Liu, J., Mao, A., Ning, G., Cao, Y., Zhang, W., and Wang, Q. (2023). Tmem88 confines ectodermal Wnt2bb signaling in pharyngeal arch artery progenitors for balancing cell cycle progression and cell fate decision. *Nature Cardiovascular Research* 2, 234-250. 10.1038/s44161-023-00215-z.
13. Pijuan-Sala, B., Griffiths, J.A., Guibentif, C., Hiscock, T.W., Jawaid, W., Calero-Nieto, F.J., Mulas, C., Ibarra-Soria, X., Tyser, R.C.V., Ho, D.L.L., et al. (2019). A single-cell molecular map of mouse gastrulation and early organogenesis. *Nature* 566, 490-495. 10.1038/s41586-019-0933-9.
14. Sun, Y., Shim, W.J., Shen, S., Sinniah, E., Pham, D., Su, Z., Mizikovskiy, D., White, M.D., Ho, Joshua W.K., Nguyen, Q., et al. (2023). Inferring cell diversity in single cell data using consortium-scale epigenetic data as a biological anchor for cell identity. *Nucleic Acids Research*, gkad307. 10.1093/nar/gkad307.
15. Bidy, B.A., Kong, W., Kamimoto, K., Guo, C., Waye, S.E., Sun, T., and Morris, S.A. (2018). Single-cell mapping of lineage and identity in direct reprogramming. *Nature* 564, 219-224. 10.1038/s41586-018-0744-4.
16. Kong, W., Bidy, B.A., Kamimoto, K., Amrute, J.M., Butka, E.G., and Morris, S.A. (2020). CellTagging: combinatorial indexing to simultaneously map lineage and identity at single-cell resolution. *Nature Protocols* 15, 750-772. 10.1038/s41596-019-0247-2.
17. VanHorn, S., and Morris, S.A. (2020). Next-Generation Lineage Tracing and Fate Mapping to Interrogate Development. *Developmental Cell*. 10.1016/j.devcel.2020.10.021.
18. Agarwal, P., Wylie, J.N., Galceran, J., Arkhitko, O., Li, C., Deng, C., Grosschedl, R., and Bruneau, B.G. (2003). Tbx5 is essential for forelimb bud initiation following patterning of the limb field in the mouse embryo. *Development* 130, 623-633. 10.1242/dev.00191.

19. Cohen, E.D., Wang, Z., Lepore, J.J., Lu, M.M., Taketo, M.M., Epstein, D.J., and Morrisey, E.E. (2007). Wnt/beta-catenin signaling promotes expansion of Isl-1-positive cardiac progenitor cells through regulation of FGF signaling. *J Clin Invest* *117*, 1794-1804. 10.1172/jci31731.
20. Li, C., Hu, L., Xiao, J., Chen, H., Li, J.T., Bellusci, S., Delanghe, S., and Minoo, P. (2005). Wnt5a regulates Shh and Fgf10 signaling during lung development. *Dev Biol* *287*, 86-97. 10.1016/j.ydbio.2005.08.035.
21. Volckaert, T., and De Langhe, S.P. (2015). Wnt and FGF mediated epithelial-mesenchymal crosstalk during lung development. *Dev Dyn* *244*, 342-366. 10.1002/dvdy.24234.
22. Guo, C., Kong, W., Kamimoto, K., Rivera-Gonzalez, G.C., Yang, X., Kirita, Y., and Morris, S.A. (2019). CellTag Indexing: genetic barcode-based sample multiplexing for single-cell genomics. *Genome Biology* *20*, 90. 10.1186/s13059-019-1699-y.
23. Qian, K., Huang, C.L., Chen, H., Blackbourn, L.W.I.V., Chen, Y., Cao, J., Yao, L., Sauvey, C., Du, Z., and Zhang, S.C. (2014). A Simple and Efficient System for Regulating Gene Expression in Human Pluripotent Stem Cells and Derivatives. *Stem Cells* *32*, 1230-1238. 10.1002/stem.1653.
24. Oceguera-Yanez, F., Kim, S.-I., Matsumoto, T., Tan, G.W., Xiang, L., Hatani, T., Kondo, T., Ikeya, M., Yoshida, Y., Inoue, H., and Woltjen, K. (2016). Engineering the AAVS1 locus for consistent and scalable transgene expression in human iPSCs and their differentiated derivatives. *Methods* *101*, 43-55. <https://doi.org/10.1016/j.ymeth.2015.12.012>.
25. Palpant, N.J., Pabon, L., Rabinowitz, J.S., Hadland, B.K., Stoick-Cooper, C.L., Paige, S.L., Bernstein, I.D., Moon, R.T., and Murry, C.E. (2013). Transmembrane protein 88: a Wnt regulatory protein that specifies cardiomyocyte development. *Development* *140*, 3799-3808. 10.1242/dev.094789.

REVIEWERS' COMMENTS

Reviewer #1 (Remarks to the Author):

The authors have satisfactorily responded to our comments. They have included SMR analysis for the other 3 high-scoring WNT-related genes. They have also made the wording/phrasing changes to interpret their results more carefully, and updated the figures to convey their points better. For the inhibition and promotion of BMP signaling leading to the same cell types, they have hypothesized that the transient nature of BMP inhibition between days 3 and 5 could cause the same cell types in the BMP promotion and inhibition groups at day 9. They mention that testing this would require substantial follow-up work, but they have revised the wording in the revised manuscript to address this, which is acceptable for the scope of this paper. The TRIAGE tools and analyses are also explained better in the text now. They have also adjusted the text to convey the fine tuning role of Tmem88 KO and pointed to similar studies that corroborate this finding. Overall, we can recommend publication of this manuscript.

Reviewer #2 (Remarks to the Author):

The authors have addressed all my concerns in response to reviewer file. The revised manuscript has improved significantly. I appreciate the great efforts made by the team and support to accept this manuscript.

Reviewer #3 (Remarks to the Author):

The authors fail to address key issues raised by the Reviewer, mainly the significance and novelty of their findings. Reiterating the previous review, the authors have already generated an in vitro cell atlas that is similar to others in the field, including from the senior author (PMID: 30290179). They also fail to address comparing their data to other in vitro and in vivo atlases as noted in previous review. Finally, it has been reported and published on several occasions that TMEM88 is involved in cardiovascular development through modulating Wnt signaling. These reports include publications in hESCs over 10 years ago (PMID: 23924634) by the senior author while in the Murry lab. Thus, major issues raised in the previous reviews have been somewhat ignored and not addressed. Additionally, in the TMEM88 KD, cardiomyocytes are unchanged at day 9 (Figure 5g) by scRNA-seq, but are significantly different at day 15 by flow cytometry (S5f), so there is a discrepancy within the data that raises concern. Also, the authors tried to address the mild blood pressure phenotype by

adding micro-CT data, but now their strongest phenotype is an increase in brain size rather than blood pressure which is not particularly consistent with their overall conclusions (Figure S5h). Overall, the manuscript may be more suitable for a specialty journal as the current manuscript remains derivative of previous work including those from the authors. I would expect that for publication in Nature communication which has a broader audience, the authors would report more novel, significant and rigorous findings than what is currently reported.

Reviewer #4 (Remarks to the Author):

Reviewer #5 (Remarks to the Author):

Reviewer #3 (Remarks to the Author):

The authors fail to address key issues raised by the Reviewer, mainly the significance and novelty of their findings. Reiterating the previous review, the authors have already generated an *in vitro* cell atlas that is similar to others in the field, including from the senior author (PMID: 30290179).

Since submission of the original manuscript, we have provided new data and/or revisions for every recommendation requested by all reviewers and refer to supportive comments from Reviewers 1 and 2 regarding the novelty of the revised study. Additionally, the cell atlas mentioned by the Reviewer (Friedman et al., 2018) differs significantly in both resolution and scope from the one presented in this manuscript. Friedman et al., (2018) include only five time points (days 0, 2, 5, 15, 30) across a cardiomyocyte-specific differentiation protocol. Our dataset focuses on multilineage differentiation between days 2 and 9 involving analysis of cell differentiation decisions across smaller time intervals and in response to developmental signalling perturbagens.

They also fail to address comparing their data to other *in vitro* and *in vivo* atlases as noted in previous review.

We benchmarked our dataset against *in vivo* reference data to achieve meaningful annotation results. In contrast to the Reviewer's comment, we evaluated our dataset against five different *in vivo* datasets (**Figure 3b-d, Supplementary Figure 2d, Supplementary Figure 3c**). In the previous review, Reviewer 3 did not request any comparisons of our dataset to any *in vitro* or *in vivo* atlases. Reviewer 2 requested an additional comparison to an *in vivo* mouse dataset by Qiu et al. (2024), which we provided in **Supplementary Figure 2d**, to their satisfaction.

Finally, it has been reported and published on several occasions that *TMEM88* is involved in cardiovascular development through modulating Wnt signaling. These reports include publications in hESCs over 10 years ago (PMID: 23924634) by the senior author while in the Murry lab. Thus, major issues raised in the previous reviews have been somewhat ignored and not addressed.

The Reviewer is correct that previous studies on *TMEM88* have focused on understanding its role in heart development. The current study has significantly expanded on this knowledge by providing a genetic loss of function analysis of *TMEM88* across diverse cell lineages and tissues *in vitro* and *in vivo*. Most importantly, the findings around *TMEM88* in this study are intended to illustrate a more impactful conceptual advance involving our stepwise approach for linking multilineage *in vitro* differentiation and human complex trait genetics to identify novel genetic regulators of developmental physiology.

Additionally, in the *TMEM88* KD, cardiomyocytes are unchanged at day 9 (Figure 5g) by scRNA-seq but are significantly different at day 15 by flow cytometry (S5f), so there is a discrepancy within the data that raises concern.

The cell type proportions shown in **Figure 5g** are derived from the TRIAGE-Cluster peaks, which do not capture all cardiomyocytes (compared to total cells surrounding the cell type peak 9 in **Figure 5d**). This is a limitation of TRIAGE-Cluster that we have previously acknowledged (Sun et al, Nucleic Acids Research, 2023). Whole-dataset clustering, and evaluation of expression of cardiomyocyte markers between the samples does demonstrate a loss in cardiomyocytes with *TMEM88* KD. As the Reviewer mentions, this effect on cardiomyocyte differentiation has already been reported, therefore justifying the focus in this manuscript on understand the role of *TMEM88* on non-cardiac lineage differentiation and organ system development.

Also, the authors tried to address the mild blood pressure phenotype by adding micro-CT data, but now their strongest phenotype is an increase in brain size rather than blood pressure which is not particularly consistent with their overall conclusions (Figure S5h).

The Reviewer has misinterpreted the brain size phenotype in **Figure S6h** (previously Figure S5h). The difference in brain size has equivalent statistical significance to the other traits shown (exact p-values now provided in the figure), the larger effect size (y-axis) is only due to a greater organ-body volume ratio for the brain compared to the other, smaller organs.